# The Breakdown of Gaussian Universality in Classification of High-dimensional Linear Factor Mixtures

**Xiaoyi Mai**
Institut de Mathématiques de Toulouse (IMT)
University of Toulouse-Jean Jaurès
Toulouse, France
xiaoyi.mai@math.univ-toulouse.fr

**Zhenyu Liao**[*]
EIC
Huazhong University of Science & Technology
Wuhan, China
zhenyu_liao@hust.edu.cn

## Abstract

The assumption of Gaussian or Gaussian mixture data has been extensively exploited in a long series of precise performance analyses of machine learning (ML) methods, on large datasets having comparably numerous samples and features. To relax this restrictive assumption, subsequent efforts have been devoted to establish "Gaussian equivalent principles" by studying scenarios of Gaussian universality where the asymptotic performance of ML methods on non-Gaussian data remains unchanged when replaced with Gaussian data having the *same mean and covariance*. Beyond the realm of Gaussian universality, there are few exact results on how the data distribution affects the learning performance.

In this article, we provide a precise high-dimensional characterization of empirical risk minimization, for classification under a general mixture data setting of *linear factor models* that extends Gaussian mixtures. The Gaussian universality is shown to break down under this setting, in the sense that the asymptotic learning performance depends on the data distribution *beyond* the class means and covariances. To clarify the limitations of Gaussian universality in the classification of mixture data and to understand the impact of its breakdown, we specify conditions for Gaussian universality and discuss their implications for the choice of loss function.

## 1 Introduction

Modern machine learning (ML) is dealing with increasingly larger datasets having high-dimensional features, using large-scale models of increasing complexity. Understanding the generalization ability of these large-scale ML models has become a major focus of research efforts (Bartlett et al., 2020; Loog et al., 2020; Nakkiran et al., 2021). One analysis approach of growing popularity is the high-dimensional asymptotic analysis (Liao & Couillet, 2019; Taheri et al., 2021a; Celentano & Montanari, 2022; Hastie et al., 2022; Loureiro et al., 2022; Celentano et al., 2023), by considering the regime where the number $n$ of samples and the dimension $p$ of feature vectors are commensurately large. Despite its asymptotic nature, this approach turns out to be surprisingly effective in explaining and predicting modern ML practice: the asymptotic performance curves are repetitively observed to closely match the average empirical performance curves on realistic datasets of moderate size and dimension (Couillet & Liao, 2022), and are particularly *different* from those offered by, e.g., classical maximum likelihood theory (Bean et al., 2013; Sur & Candès, 2019; Taheri et al., 2021b).

To analytically characterize the generalization performance of large-scale ML models in the aforementioned high-dimensional regime, advanced statistical tools such as the approximate message passing (Donoho & Montanari, 2016; Barbier et al., 2019), convex Gaussian min-max theorem (Thrampoulidis et al., 2018; Salehi et al., 2019; Deng et al., 2022; Javanmard & Soltanolkotabi, 2022), replica method (Huang, 2017; Gerace et al., 2020; Maillard et al., 2020), and random matrix theory (RMT) (Couillet & Liao, 2022; Mai et al., 2019; Mai & Couillet, 2021) have been carefully adapted to take into account of the nonlinearity of ML models. As these tools apply directly on

---

[*]Corresponding author: Zhenyu Liao.

Gaussian data, a majority of high-dimensional asymptotic analyses are performed under Gaussian data models in the context of regression (El Karoui et al., 2013; Donoho & Montanari, 2016; Taheri et al., 2021a; Celentano & Montanari, 2022) or Gaussian mixture models (GMMs) in the context of classification (Mignacco et al., 2020; Thrampoulidis et al., 2020; Refinetti et al., 2021).

Despite this seemingly restrictive assumption of data Gaussianity, the derived high-dimensional asymptotic results have been empirically observed to remain valid on non-Gaussian data, including both synthetic non-Gaussian data and samples drawn from realistic (say image) datasets (Loureiro et al., 2021; Taheri et al., 2021b), hinting at a phenomenon of *Gaussian universality*. This motivated a series of recent works establishing, through, e.g., an one-directional central limit theorem (CLT) argument, a Gaussian equivalent principal (GEP) for high-dimensional ML models ranging from generalized linear models to shallow neural networks (Gerace et al., 2020; Goldt et al., 2022; Hu & Lu, 2022; Montanari & Saeed, 2022; Schröder et al., 2023; Han & Shen, 2023). According to the GEP, the performance of ML methods on non-Gaussian data is asymptotically the same under an equivalent Gaussian model with matching first- and second-order moments. Assuming a conditional one-directional CLT, Dandi et al. (2024) put forward a conditional Gaussian equivalent principle (CGEP) stating that the asymptotic classification error for non-Gaussian mixtures remains unchanged when replaced by a Gaussian mixture model with identical class-conditional means and covariances. This contribution, however, did not specify the conditions required on the mixture data model for this conditional one-directional CLT to hold.

This work is driven by the need to investigate the applicability of CGEP under mixture models and to characterize the impact of non-Gaussian data variations when the CGEP does *not* apply. By considering a more general mixture model (see Definition 1) where classes are described by linear factor models—a fundamental probabilistic framework in statistical inference and generative models (Goodfellow et al., 2016, Chapter 13), our analysis extends a long line of high-dimensional asymptotic performance analyses in classification of Gaussian mixtures (Dobriban & Wager, 2018; Huang, 2017; Liao & Couillet, 2019; Mai & Liao, 2019; Huang & Yang, 2021; Kammoun & Alouini, 2021; Wang & Thrampoulidis, 2021; Pesce et al., 2023). We discuss the validity of CGEP under this linear factor mixture model and specify its conditions. On a technical level, we develop a flexible "leave-one-out" analysis approach, in a similar spirit to the analysis of robust linear regression by El Karoui et al. (2013). The elementary nature of this leave-one-out procedure allows us to extend the approach of high-dimensional asymptotic analysis beyond the realm of Gaussian universality.

**Our Contributions.** The main findings of this paper are summarized below.

1. We provide in Theorem 1 an asymptotic characterization of ridge-regularized empirical risk minimization (ERM) for classification of data drawn from a linear factor mixture model (LFMM, see Definition 1 below, that generalizes the GMM). This precise characterization gives access to the asymptotic performance on mixture data *beyond Gaussian universality*.

2. Based on the proposed analysis, we study Gaussian universality in the sense of CGEP and provide conditions on LFMM under which the data distribution affects the asymptotic learning behavior *only* via its first two class-conditional moments.

   - We first discuss in Section 5.1 the Gaussian universality on *in-distribution performance* and conclude in Corollary 2 that the training and generalization performances of ERM under a given LFMM remain unchanged under its equivalent GMM (with identical class means and covariances, see Definition 2), if all *informative factors* of the LFMM that significantly correlated with the class label are *class-conditional normal variables*.

   - We then investigate in Section 5.2 the Gaussian universality of *classifier* (see Definition 3) and conclude in Corollary 3 that on a given test set (of arbitrary distribution), the ERM classifier trained on data drawn from an LFMM gives the same asymptotic classification error as the one trained on its equivalent GMM, whenever the square loss is used and/or in the case of class-conditional Gaussian informative factors for LFMM.

3. While it has been known that for high-dimensional GMM, the square loss is optimal in both unregularized (Taheri et al., 2021b) and ridge-regularized (Mai & Liao, 2019) classifications, it is *no longer* the case under the general LFMM due to the breakdown of Gaussian universality. We discuss in Section 5.2 how the suboptimality of square loss under LFMM relates to its effect on the Gaussian universality of the ERM classifier. Our analysis thus opens the door to future investigation on the optimal loss design for *non-Gaussian* data.

## 2 BACKGROUND ON GAUSSIAN UNIVERSALITY IN HIGH DIMENSIONS

The Gaussian universality phenomenon was observed in many high-dimensional inference or ML problems, where some key statistics such as estimation error or classification accuracy exhibit universal behaviors independent of the data distribution. This phenomenon was extensively exploited to relax the data Gaussianity assumption that underlined many results in high-dimensional statistics, through a universality argument often established with two key ingredients - the law of large numbers (LLN) and the CLT.

Here we briefly review previous findings on Gaussian universality in the high-dimensional regime.

**Universality of large sample covariance matrices.** It has been long known in RMT that the eigenspectra of large random matrices enjoy universal properties for Gaussian and non-Gaussian entries (Tao et al., 2010; Pastur & Shcherbina, 2011). Fundamentally, Marchenko & Pastur (1967) put forward that for sample covariance matrices of the type $\frac{1}{n}\sum_{i=1}^{n} \mathbf{x}_i\mathbf{x}_i^\mathsf{T} \in \mathbb{R}^{p \times p}$ obtained from $n$ i.i.d. data vectors $\mathbf{x}_i$ of dimension $p$, the universality on the limiting eigenvalue distribution hinges on the concentration of quadratic forms of $\mathbf{x}_i$ around their expectations, that is

$$\lim_{n,p \to \infty} (\mathbf{x}_i^\mathsf{T}\mathbf{M}\mathbf{x}_i - \mathbb{E}[\mathbf{x}_i^\mathsf{T}\mathbf{M}\mathbf{x}_i])/\mathbb{E}[\mathbf{x}_i^\mathsf{T}\mathbf{M}\mathbf{x}_i] = 0, \tag{1}$$

for deterministic $\mathbf{M} \in \mathbb{R}^{p \times p}$. This LLN-type result holds for a wide family of high-dimensional random vectors $\mathbf{x}_i$. An important example studied by Bai & Silverstein (2008) is $\mathbf{x}_i = \mathbf{\Sigma}^{\frac{1}{2}}\mathbf{z}_i$ with $\mathbf{z}_i$ of i.i.d. standardized entries with bounded fourth moments and non-negative definite symmetric $\mathbf{\Sigma}$.

**Universality of empirical risk minimization.** In line with the universal behavior of large sample covariance matrices, it has been recently demonstrated in a series of works (Gerace et al., 2020; Goldt et al., 2022; Hu & Lu, 2022; Montanari & Saeed, 2022; Schröder et al., 2023) that random (and deterministic under certain conditions) feature maps can produce feature matrices that, when replaced by "equivalent" Gaussian features with the same first- and second-order moments, yield the same training and/or generalization performance for many ML methods. This GEP has also been proven for data vectors of independent entries in the context of regularized regression (Montanari & Nguyen, 2017; Panahi & Hassibi, 2017; Han & Shen, 2023).

In the context of ERM, the GEP traced back to a CLT on the inner product $\mathbf{x}^\mathsf{T}\boldsymbol{\beta}$ for feature vector $\mathbf{x} \in \mathbb{R}^p$ independent of classifier $\boldsymbol{\beta}$ living in a subspace $\mathcal{B} \subset \mathbb{R}^p$ containing the ERM solution $\hat{\boldsymbol{\beta}}$, i.e.,

$$\lim_{n,p \to \infty} \sup_{\boldsymbol{\beta} \in \mathcal{B}} \left( \mathbb{E}[f(\mathbf{x}^\mathsf{T}\boldsymbol{\beta})] - \mathbb{E}[f(\mathbf{g}^\mathsf{T}\boldsymbol{\beta})] \right) = 0, \tag{2}$$

with $\mathbf{g} \sim \mathcal{N}(\mathbb{E}[\mathbf{x}], \mathrm{Cov}[\mathbf{x}])$ being the "equivalent" Gaussian vector, for any bounded Lipschitz function $f \colon \mathbb{R} \to \mathbb{R}$. The one-directional CLT in (2) requires the ERM solution $\hat{\boldsymbol{\beta}}$ to not particularly aligned with any non-Gaussian variation in the feature vector $\mathbf{x}$, in order to ensure the asymptotic normality of $\mathbf{x}^\mathsf{T}\boldsymbol{\beta}$ per a CLT-type argument.

**Universality of empirical risk minimization on mixture data.** Inspired by the one-directional CLT condition (2) for GEP in ERM, Dandi et al. (2024) demonstrated the Gaussian universality for mixture models under a key assumption that is a conditional version of (2):

$$\lim_{n,p \to \infty} \sup_{\boldsymbol{\beta} \in \mathcal{B}} \left( \mathbb{E}\left[ f(\mathbf{x}^\mathsf{T}\boldsymbol{\beta}) | y_\mathbf{x} = C \right] - \mathbb{E}\left[ f(\mathbf{g}_{[C]}^\mathsf{T}\boldsymbol{\beta}) \right] \right) = 0, \tag{3}$$

where $y_\mathbf{x}$ is the class label of $\mathbf{x}$ , $C$ a class modality, and $\mathbf{g}_{[C]} \sim \mathcal{N}(\mathbb{E}[\mathbf{x}|y_\mathbf{x} = C], \mathrm{Cov}[\mathbf{x}|y_\mathbf{x} = C])$. Under this conditional one-directional CLT in (3), Dandi et al. (2024) showed that the asymptotic training and generalization errors only depend on the class-conditional means and covariances of the mixture model, obeying thus a conditional Gaussian equivalent principle (CGEP).

However, the condition in (3) is *not* verifiable from the data distribution, making it essentially different from the ones given in Section 5.

Other related works established equivalences between Gaussian data and Gaussian mixtures. For classification with random labels $y_\mathbf{x} \sim \mathrm{Unif}(\{-1, 1\})$ generated independently of $\mathbf{x}$, Gerace et al. (2024) proved that the training loss on GMM is asymptotically equal to that on a *single* Gaussian. Pesce et al. (2023) considered a teacher-student model and showed that when the target label $y$ is generated by a teacher model *uncorrelated* with cluster means, the same asymptotic performance can be obtained by replacing a homoscedastic (i.e., having identical covariance) Gaussian mixture with a single Gaussian.

**Universality under elliptical distributions.** For "elliptic-like" data vectors of form $\mathbf{x} = a\mathbf{M}\mathbf{u}$ with $a \in \mathbb{R}$ a random scaling variable, $\mathbf{M} \in \mathbb{R}^{p \times d}$ a deterministic matrix and $\mathbf{u} \in \mathbb{R}^d$ a vector of standardized variables satisfying the concentration of quadratic forms in (1) (e.g., $\mathbf{u} \sim \mathcal{N}(\mathbf{0}_d, \mathbf{I}_d)$), El Karoui (2009) revealed a universal limiting spectrum for the sample covariance matrix that is insensitive to the distribution of $\mathbf{u}$ but depends on the scaling variable $a$.

Due to the existence of the scaling variable $a$, the one-directional CLT in (2) can not hold unless when conditioned on $a$. This remark was confirmed by the findings of El Karoui (2018); Adomaityte et al. (2024). For $\mathbf{M} = \mathbf{I}_p$ and $\mathbf{u}$ of i.i.d. entries, El Karoui (2018) characterized the asymptotic error of ridge-regularized regression, which is universal with respect to the distribution of $\mathbf{u}$ but not $a$. In other words, the GEP collapses while the CGEP with respect to the scaling factor $a$ can still apply in this setting. Adomaityte et al. (2024) considered a mixture model $\mathbf{x} \sim \mathcal{N}(y\boldsymbol{\mu}, a\mathbf{I}_p)$ with label $y = \pm 1$ and random scaling factor $a$, under which the asymptotic classification error is *non-universal* with respect to the distribution of $a$. Here with our analysis under LFMM, we show that Gaussian universality may breakdown even for data vectors of concentrated quadratic forms as described in (1), a condition not satisfied by elliptical data vectors due to the presence of a random scaling factor $a$.

## 3 PROBLEM SETUP

For a set of $n$ training samples $\{(\mathbf{x}_i, y_i)\}_{i=1}^n$ with feature vectors $\mathbf{x}_i \in \mathbb{R}^p$ and binary labels $y_i \in \{\pm 1\}$, a classifier is trained by minimizing the following ridge-regularized empirical risk:

$$\hat{\boldsymbol{\beta}}_{\ell,\lambda} = \arg\min_{\boldsymbol{\beta} \in \mathbb{R}^p} \frac{1}{n} \sum_{i=1}^n \ell(\mathbf{x}_i^\mathsf{T} \boldsymbol{\beta}, y_i) + \frac{\lambda}{2} \|\boldsymbol{\beta}\|^2, \tag{4}$$

for some non-negative loss function $\ell \colon \mathbb{R} \times \{\pm 1\} \to \mathbb{R}_+$ that evaluates the difference between the classification score $\hat{y}_i = \boldsymbol{\beta}^\mathsf{T} \mathbf{x}_i$ and the corresponding ground-truth label $y_i$. Data instances $\mathbf{x}$ with negative scores $\boldsymbol{\beta}^\mathsf{T} \mathbf{x}$ will be assigned to the class of label $y = -1$, and those with positive scores to the class annotated by $y = 1$. The addition of the $l_2$ regularization term with $\lambda > 0$ can improve the generalization through a better bias-variance trade-off, and also ensures the uniqueness of the solution $\hat{\boldsymbol{\beta}}_{\ell,\lambda}$ in the over-parametrized regime where the feature dimension $p$ is greater than the sample size $n$.

In this paper, we consider convex and continuously differentiable loss functions.

**Assumption 1** (Loss function). *The function $\ell(\cdot, y) \colon \mathbb{R} \to \mathbb{R}_+$ in (4) is convex and continuously differentiable with its (first) derivative different from $0$ at the origin. Its second and third derivatives exist and are bounded, except on a finite set of points.*

Assumption 1 holds for the logistic loss $\ell(\hat{y}, y) = -\ln(1/(1 + e^{-y\hat{y}}))$ used in logistic regression, the square loss $\ell(\hat{y}, y) = (y - \hat{y})^2/2$ for least-squares classifier, and the square hinge loss $\ell(\hat{y}, y) = \max\{0, 1 - y\hat{y}\}^2$. Non-smooth losses such as the hinge loss $\ell(\hat{y}, y) = \max\{0, 1 - y\hat{y}\}$ used in SVMs (Schölkopf & Smola, 2018), and the absolute loss $\ell(\hat{y}, y) = |\hat{y} - y|$, fail to meet Assumption 1.[1]

In the following, we focus on the ERM in (4), and use the shorthand notation $\hat{\boldsymbol{\beta}}$ for $\hat{\boldsymbol{\beta}}_{\ell,\lambda}$ in (4) unless there is a risk of confusion. We consider the following linear factor mixture model.

**Definition 1** (Linear factor mixture model, LFMM). *We say that a data instance $(\mathbf{x}, y) \sim \mathcal{D}_{(\mathbf{x},y)}$ with class label $y \in \{\pm 1\}$ and class priors $\Pr(y = -1) = \rho$, $\Pr(y = 1) = 1 - \rho$, follows a linear factor mixture model if the corresponding feature vector $\mathbf{x} \in \mathbb{R}^p$ can be expressed as a linear mapping of $p$ independent factors $z_1, \ldots, z_p$ as*

$$\mathbf{x} = \sum_{k=1}^p z_k \mathbf{v}_k = \sum_{k=1}^p (y s_k + e_k) \mathbf{v}_k, \tag{5}$$

*for linearly independent deterministic vectors $\mathbf{v}_1, \ldots, \mathbf{v}_p \in \mathbb{R}^p$ and standardized independent[2] noises $e_1, \ldots, e_p \in \mathbb{R}$ of symmetric distribution. Among the $p$ factors $z_1, \ldots, z_p$, we have*

- *$q$ **informative factors** $z_1, \ldots, z_q$ with deterministic signals $s_k > 0$, $\forall k \in \{1, \ldots, q\}$; and*

- *$p - q$ **noise factors** $z_{q+1}, \ldots, z_p$ with $s_k = 0$, $\forall k \in \{q+1, \ldots, p\}$.*

---

[1]To study these non-smooth losses, a workaround would be to evaluate instead a series of smooth functions that gradually approach the non-smooth functions, so as to retrieve their performance in some carefully taken limit. Such consideration is however beyond the focus of this paper.

[2]In other words, $\mathbb{E}[e_k] = 0$, $\text{Var}[e_k] = 1$, $\forall k \in \{1, \ldots, p\}$.

*Note that* (5) *can be compactly written as* $\mathbf{x} = \mathbf{V}\mathbf{z}$, *with* $\mathbf{V} = [\mathbf{v}_1, \ldots, \mathbf{v}_p] \in \mathbb{R}^{p \times p}$ *and* $\mathbf{z} = [z_1, \ldots, z_p]^\mathsf{T} = [ys_1 + e_1, \ldots, ys_q + e_q, e_{q+1}, \ldots, e_p]^\mathsf{T} \in \mathbb{R}^p$. *The class-conditional means and covariances of* $\mathbf{x}$ *are therefore given by*

$$\boldsymbol{\mu} \equiv \mathbb{E}[\mathbf{x}|y=1] = \textstyle\sum_{k=1}^q s_k \mathbf{v}_k \in \mathbb{R}^p, \quad \mathbb{E}[\mathbf{x}|y=-1] = -\boldsymbol{\mu}, \tag{6}$$

$$\boldsymbol{\Sigma} \equiv \mathrm{Cov}[\mathbf{x}|y=\pm 1] = \mathbf{V}\mathbf{V}^\mathsf{T} = \textstyle\sum_{k=1}^p \mathbf{v}_k \mathbf{v}_k^\mathsf{T} \in \mathbb{R}^{p \times p}. \tag{7}$$

Notice that GMM of form $\mathbf{x} \sim \mathcal{N}(y\boldsymbol{\mu}, \boldsymbol{\Sigma})$ is a special case of LFMM in Definition 1 with exclusively Gaussian noises $e_1, \ldots, e_p$. See also Definition 2 below for the associated equivalent GMM.

Linear factor models are among the most fundamental probabilistic models with latent variables, which underlie many ML methods such as PCA and ICA, and serve as building blocks of deep generative models (Goodfellow et al., 2016, Chapter 13). They are often expressed as:

$$\mathbf{x} = \mathbf{W}\mathbf{h} + \mathbf{b} + \mathrm{noise},$$

where $\mathbf{h}$ is a vector of latent variables, $\mathbf{b}$ a constant bias, and noise stands for an uninformative term of independent Gaussian noises. The LFMM in Definition 1 can be related to this form minus the bias $\mathbf{b}$. Our framework requires the clusters to have opposite means (therefore $\mathbf{b} = \mathbf{0}_p$), which can be satisfied through a centering operation on the original data space.

Our analysis applies under the following assumption on the distribution of LFMM.

**Assumption 2** (Distribution of LFMM). *We consider, for the LFMM in Definition 1, that (i) the factors* $z_1, \ldots, z_p$ *have bounded fourth moments and (ii) the signal subspace* $\mathrm{Span}\{\mathbf{v}_1, \ldots, \mathbf{v}_q\}$ *is orthogonal to the noise subspace* $\mathrm{Span}\{\mathbf{v}_{q+1}, \ldots, \mathbf{v}_p\}$.

The condition of bounded fourth moment for $z_1, \ldots, z_p$ in Item (i) of Assumption 2 is required for some concentration results in our high-dimensional asymptotic analysis and Item (ii) separates the informative signal subspace from the noise subspace (in which no classifier can achieve better performance than random guesses).

We position ourselves under the following high-dimensional asymptotic setting, where the feature dimension $p$ and sample size $n$ are both large and comparable.

**Assumption 3** (High-dimensional regime). *As* $n \to \infty$ *with fixed* $n/p \in (0, \infty)$, *we have, for the LFMM in Definition 1 that (i)* $\|\boldsymbol{\mu}\|, \|\boldsymbol{\Sigma}\|, \|\boldsymbol{\Sigma}^{-1}\| = \Theta(1)$ *and (ii)* $s_1, \ldots, s_q = \Theta(1)$ *with fixed* $q$.

In plain words, Assumption 3 says that the ratio $n/p$, or the number of samples per dimension, remains finite in high dimensions. Item (i) of Assumption 3 ensures, by bounding $\|\boldsymbol{\mu}\|$ and $\|\boldsymbol{\mu}\|^{-1}$, that the distance between the LFMM class centers is comparable to 1. It also guarantees, by controlling $\|\boldsymbol{\Sigma}\|$ and $\|\boldsymbol{\Sigma}^{-1}\|$, that the variation of feature vector $\mathbf{x}$ on any direction in $\mathbb{R}^p$ is also comparable to 1. This implies that the feature vector $\mathbf{x}$ does not live in a subspace of dimension smaller than $p$. The fixed number $q$ of informative factors in Item (ii) of Assumption 3 is a consequence of $\|\boldsymbol{\mu}\| = \|\sum_{k=1}^q s_k \mathbf{v}_k\| = \Theta(1)$.

## 4 HIGH-DIMENSIONAL ASYMPTOTIC PERFORMANCE UNDER LFMM

In this section, we present a self-consistent system of equations that gives access to the high-dimensional training and generalization performances of the ERM classifier in (4), under the LFMM in Definition 1. The characterization of high-dimensional asymptotic performance via a system of equations is reminiscent of previous analyses under GMM (Mai & Liao, 2019; Mignacco et al., 2020; Pesce et al., 2023), but our equations are different due to the collapse of the conditional one-dimensional CLT in (3) required for applying the CGEP.

Before presenting our system of equations, let us introduce first some mathematical objects involved in these equations. With the proximal operator $\mathrm{prox}_{\tau, f}(t) = \arg\min_{a \in \mathbb{R}} \left[ f(a) + \frac{1}{2\tau}(a-t)^2 \right]$ for $\tau > 0$ and convex $f: \mathbb{R} \to \mathbb{R}$, we define the mapping

$$h_\kappa(t, y) = (\mathrm{prox}_{\kappa, \ell(\cdot, y)}(t) - t)/\kappa, \tag{8}$$

for some constant $\kappa > 0$. Let $r \in \mathbb{R}$ be a random variable of form

$$r = ym + \sigma\tilde{e} + \textstyle\sum_{k=1}^q \psi_k e_k, \tag{9}$$

for constants $m, \sigma, \psi_1, \ldots, \psi_q$, with label $y$ and $e_1, \ldots, e_q$ the corresponding noise variables in the informative factors $z_1, \ldots, z_q$ of the LFMM in Definition 1, as well as $\tilde{e} \sim \mathcal{N}(0,1)$ independent of $y, z_1, \ldots, z_q$. Remark that the distribution of $r$ is parameterized by $m, \sigma^2, \psi_1, \ldots, \psi_q$.

**Self-consistent system of equations.** Our system of equations is on the $q+3$ deterministic constants $\theta, \eta, \gamma, \omega_1, \ldots, \omega_q$ that fully characterize the asymptotic performance of ERM classifier trained on high-dimensional LFMM[3]:

$$\theta = -\mathbb{E}\left[\frac{\partial h_\kappa(r,y)}{\partial r}\right], \quad \eta = \mathbb{E}[y h_\kappa(r,y)], \quad \gamma = \sqrt{\mathbb{E}[h_\kappa^2(r,y)]},$$

$$\omega_k = \mathbb{E}[h_\kappa(r,y)e_k] + \theta \cdot \mathbf{v}_k^\mathsf{T} \mathbf{Q} \boldsymbol{\xi}, \quad \forall k \in \{1, \ldots, q\}, \tag{10}$$

where

$$\boldsymbol{\xi} = \eta \boldsymbol{\mu} + \textstyle\sum_{k=1}^q \omega_k \mathbf{v}_k, \quad \mathbf{Q} = (\lambda \mathbf{I}_p + \theta \boldsymbol{\Sigma})^{-1}, \tag{11}$$

the mapping $h_\kappa(r,y)$ is as defined in (8) for

$$\kappa = \tfrac{1}{n} \operatorname{tr} \boldsymbol{\Sigma} \mathbf{Q}, \tag{12}$$

and the random variable $r$ as defined in (9) with

$$m = \boldsymbol{\mu}^\mathsf{T} \mathbf{Q} \boldsymbol{\xi}, \quad \sigma^2 = \tfrac{\gamma^2}{n} \operatorname{tr}(\mathbf{Q}\boldsymbol{\Sigma})^2, \quad \psi_k = \mathbf{v}_k^\mathsf{T} \mathbf{Q} \boldsymbol{\xi}, \quad \forall k \in \{1, \ldots, q\}. \tag{13}$$

We are now ready to present our Theorem 1 on the asymptotic distributions of in-sample and out-of-sample predicted scores. The proof of Theorem 1 is provided in Appendix A.1.

**Theorem 1** (Asymptotic distribution of predicted scores). *Let Assumptions 1, 2, and 3 hold, for $\hat{\boldsymbol{\beta}}$ solution to the ERM problem in (4) on a training set $\{(\mathbf{x}_i, y_i)\}_{i=1}^n$ of size $n$ drawn i.i.d. $(\mathbf{x}_i, y_i) \sim \mathcal{D}_{(\mathbf{x},y)}$ from the LFMM in Definition 1, we have that, for any bounded Lipschitz function $f: \mathbb{R} \to \mathbb{R}$,*

$$\mathbb{E}\left[f(\hat{\boldsymbol{\beta}}^\mathsf{T} \boldsymbol{\nu})\right] - \mathbb{E}\left[f(\tilde{\boldsymbol{\beta}}^\mathsf{T} \boldsymbol{\nu})\right] \to 0, \tag{14}$$

*for any deterministic feature vector $\boldsymbol{\nu} \in \mathbb{R}^p$, and*

$$\mathbb{E}[f(\hat{\boldsymbol{\beta}}^\mathsf{T} \mathbf{x}_i)] - \mathbb{E}[f(\operatorname{prox}_{\kappa, \ell(\cdot, y_i)}(\tilde{\boldsymbol{\beta}}^\mathsf{T} \mathbf{x}_i))] \to 0, \quad \forall i \in \{1, \ldots, n\}, \tag{15}$$

*where*

$$\tilde{\boldsymbol{\beta}} = (\lambda \mathbf{I}_p + \theta \boldsymbol{\Sigma})^{-1} \left(\eta \boldsymbol{\mu} + \textstyle\sum_{k=1}^q \omega_k \mathbf{v}_k + \gamma \boldsymbol{\Sigma}^{\frac{1}{2}} \mathbf{u}\right), \tag{16}$$

*for Gaussian vector $\mathbf{u} \sim \mathcal{N}(\mathbf{0}_p, \mathbf{I}_p/n)$ independent of $\{(\mathbf{x}_i, y_i)\}_{i=1}^n$ and constants $\theta, \eta, \gamma, \omega_1, \ldots, \omega_q$ determined by the self-consistent system of equations in (10), with $\kappa$ given in (12).*

According to (14) in Theorem 1, for a fresh test sample $(\mathbf{x}', y')$ (which might be drawn from a distribution *different* from $\mathcal{D}_{(\mathbf{x},y)}$ of training samples, as in the case of transfer learning), the out-of-sample predicted scores $\hat{\boldsymbol{\beta}}^\mathsf{T} \mathbf{x}', \tilde{\boldsymbol{\beta}}^\mathsf{T} \mathbf{x}'$ produced by the ERM classifier $\hat{\boldsymbol{\beta}}$ and its high-dimensional "equivalent" $\tilde{\boldsymbol{\beta}}$ given in (16) have asymptotically the same distribution in the sense of (14). Furthermore, (15) tells us that the in-sample predicted score $\hat{\boldsymbol{\beta}}^\mathsf{T} \mathbf{x}_i$ of $\hat{\boldsymbol{\beta}}$ on a training sample $(\mathbf{x}_i, y_i)$ follows asymptotically the same distribution as $\operatorname{prox}_{\kappa, \ell(\cdot, y_i)}(\tilde{\boldsymbol{\beta}}^\mathsf{T} \mathbf{x}_i)$. Since the distribution of $\tilde{\boldsymbol{\beta}}$ is given in (16), we obtain directly from Theorem 1 the asymptotic training and generalization errors of the ERM classifier $\hat{\boldsymbol{\beta}}$.

Furthermore, it follows from LLN and CLT that $(\tilde{\boldsymbol{\beta}}^\mathsf{T} \mathbf{x}, y)$ with $(\mathbf{x}, y) \sim \mathcal{D}_{(\mathbf{x},y)}$ independent of $\tilde{\boldsymbol{\beta}}$ converges in distribution to $(r, y)$ with $r$ as defined in (9) with $m, \sigma^2, \psi_1, \ldots, \psi_q$ given in (13). We thus obtain the following corollary on the asymptotic classification accuracy of $\hat{\boldsymbol{\beta}}$ on any training sample $(\mathbf{x}_i, y_i)$ and test sample $(\mathbf{x}', y')$ drawn from the same distribution $\mathcal{D}_{(\mathbf{x},y)}$. The proof of Corollary 1 is deferred to Appendix A.2.1.

**Corollary 1** (Asymptotic generalization and training performances). *Under the conditions and notations of Theorem 1, we have that, for any bounded Lipschitz function $f: \mathbb{R} \to \mathbb{R}$,*

$$\mathbb{E}\left[f(\tilde{\boldsymbol{\beta}}^\mathsf{T} \mathbf{x})|y\right] - \mathbb{E}\left[f(r)|y\right] \to 0, \tag{17}$$

---

[3]According to Assumption 1, $\frac{\partial h(r,y)}{\partial r}$ exists except on a finite set of points. On those points, we use the left derivative of $h(r,y)$ with respect $r$, i.e., $\lim_{t \to r_-} (h(t,y) - h(r,y))/(t-r)$.

for $(\mathbf{x}, y) \sim \mathcal{D}_{(\mathbf{x},y)}$ independent of $\tilde{\boldsymbol{\beta}}$, where $r$ is as defined in (9) with $m, \sigma^2, \psi_1, \ldots, \psi_q$ given in (13). Consequently, we have

$$\Pr(y'\hat{\boldsymbol{\beta}}^{\mathsf{T}}\mathbf{x}' > 0) - \Pr(yr > 0) \to 0, \tag{18}$$

for some test sample $(\mathbf{x}', y') \sim \mathcal{D}_{(\mathbf{x},y)}$ independent of $\{(\mathbf{x}_i, y_i)\}_{i=1}^n$, and

$$\Pr(y_i\hat{\boldsymbol{\beta}}^{\mathsf{T}}\mathbf{x}_i > 0) - \Pr(y \operatorname{prox}_{\kappa,\ell(\cdot,y)}(r) > 0) \to 0, \quad \forall i \in \{1, \ldots, n\}. \tag{19}$$

**Remark 1** (On classifier bias under GMM and LFMM). Taking expectation on both sides of (16), we get $\mathbb{E}[\tilde{\boldsymbol{\beta}}] = (\lambda\mathbf{I}_p + \theta\boldsymbol{\Sigma})^{-1}(\eta\boldsymbol{\mu} + \sum_{k=1}^q \omega_k\mathbf{v}_k)$. It follows from (10) and Stein's lemma (Ingersoll, 1987) that $\omega_1, \ldots, \omega_q = 0$ in the case of Gaussian informative factors $z_1, \ldots, z_q$. As GMM is a special case of LFMM with exclusively Gaussian (informative and noise) factors, we have that $\tilde{\boldsymbol{\beta}}$ aligns, in expectation, with $(\lambda\mathbf{I}_p + \theta\boldsymbol{\Sigma})^{-1}\boldsymbol{\mu}$ under GMM. For non-Gaussian informative factors, we generally have non-zero $\omega_1, \ldots, \omega_q$ that account for the non-Gaussian variation in data, making $\hat{\boldsymbol{\beta}}$ more or less aligned with the directions $\mathbf{v}_1, \ldots, \mathbf{v}_q$ of the informative factors.

# 5 Conditions and Implications of Gaussian Universality

In this section, we exploit our high-dimensional asymptotic analysis in Section 4 to derive the conditions of Gaussian universality under LFMM. To discuss the Gaussian universality in classification of mixture data, we introduce the notion of equivalent Gaussian mixture model (to a given LFMM), in a similar spirit to Dandi et al. (2024).

**Definition 2** (Equivalent Gaussian mixture model). *For an LFMM $\mathcal{D}_{(\mathbf{x},y)}$ as in Definition 1, we define its equivalent Gaussian mixture model (GMM) $\mathcal{D}_{(\mathbf{g},y)}$ as the GMM with the same class-conditional means and covariances as the LFMM $\mathcal{D}_{(\mathbf{x},y)}$. Namely,*

$$\mathbf{g} \sim \mathcal{N}(y\boldsymbol{\mu}, \boldsymbol{\Sigma}), \tag{20}$$

*for $\boldsymbol{\mu}, \boldsymbol{\Sigma}$ given in (6) and (7) of Definition 1, respectively. We denote by $\hat{\boldsymbol{\beta}}^{\mathbf{g}}$ the ERM solution to (4) obtained on $n$ i.i.d. equivalent GMM samples $(\mathbf{g}_1, y_1), \ldots, (\mathbf{g}_n, y_n) \sim \mathcal{D}_{(\mathbf{g},y)}$, and similarly its high-dimensional "equivalent" $\tilde{\boldsymbol{\beta}}^{\mathbf{g}}$ as in (16) of Theorem 1.*

Notice importantly from Definition 1 that the equivalent GMM $\mathcal{D}_{(\mathbf{g},y)}$ to an LFMM $\mathcal{D}_{(\mathbf{x},y)}$ can be obtained by taking $e_1, \ldots, e_p$ of the LFMM $\mathcal{D}_{(\mathbf{x},y)}$ to be standard Gaussian variables.

We define two types of Gaussian universality considered in this paper as follows.

**Definition 3** (Gaussian universality under LFMM). *For an ERM solution $\hat{\boldsymbol{\beta}}$ obtained on a general LFMM $\mathcal{D}_{(\mathbf{x},y)}$ in Definition 1 and an ERM solution $\hat{\boldsymbol{\beta}}^{\mathbf{g}}$ obtained on the equivalent GMM in Definition 2, we say that the Gaussian universality holds*

- *on **classifier** if $\hat{\boldsymbol{\beta}}$ has asymptotically the same predictive ability as $\hat{\boldsymbol{\beta}}^{\mathbf{g}}$ on a given test set, as a consequence of their high-dimensional equivalents $\tilde{\boldsymbol{\beta}}_{\ell,\lambda}, \tilde{\boldsymbol{\beta}}_{\ell,\lambda}^{\mathbf{g}}$ provided in Theorem 1 following the* same *distribution;*

- *on **in-distribution performance** if the respective training and generalization performances under $\mathcal{D}_{(\mathbf{x},y)}$ are asymptotically the same as under $\mathcal{D}_{(\mathbf{g},y)}$, that is*

$$\Pr(y_i\mathbf{x}_i^{\mathsf{T}}\hat{\boldsymbol{\beta}} > 0) - \Pr(y_i\mathbf{g}_i^{\mathsf{T}}\hat{\boldsymbol{\beta}}^{\mathbf{g}} > 0) \to 0, \tag{21}$$

*and*

$$\Pr(y'\mathbf{x}'^{\mathsf{T}}\hat{\boldsymbol{\beta}} > 0) - \Pr(y'\mathbf{g}'^{\mathsf{T}}\hat{\boldsymbol{\beta}}^{\mathbf{g}} > 0) \to 0, \tag{22}$$

*for $(\mathbf{x}', y') \sim \mathcal{D}_{(\mathbf{x},y)}$ a test sample independent of $\{(\mathbf{x}_i, y_i)\}_{i=1}^n$, and $(\mathbf{g}', y') \sim \mathcal{D}_{(\mathbf{g},y)}$ independent of $\{(\mathbf{g}_i, y_i)\}_{i=1}^n$.*

In the following, we study first in Section 5.1 the Gaussian universality in the sense of in-distribution performance, and discuss our results with respect to the conditional one-directional CLT and the CGEP in Dandi et al. (2024). We then reveal in Section 5.2 the key role of square loss in inducing the Gaussian universality of classifier, and discuss its implication for the choice of loss function.

Throughout this section, our discussions are illustrated through numerical experiments on datasets of moderately large size, with $n, p$ only in hundreds. A close match is consistently observed between the proposed asymptotic analysis and the empirical results.

| Universality | GMM | Breakdown |
|:---:|:---:|:---:|
| 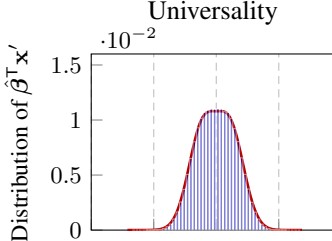 | 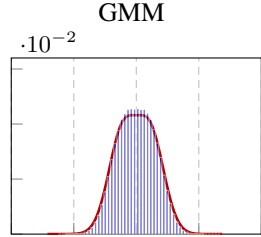 | 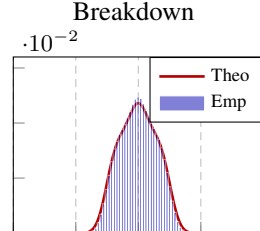 |

Figure 1: Theoretical and empirical distribution of predicted scores $\hat{\boldsymbol{\beta}}^{\mathsf{T}}\mathbf{x}'$ for some fresh $(\mathbf{x}', y') \sim \mathcal{D}_{(\mathbf{x},y)}$ independent of $\hat{\boldsymbol{\beta}}$. The theoretical probability densities (**red**) are obtained from Theorem 1, and the empirical histograms (**blue**) are the values of $\hat{\boldsymbol{\beta}}^{\mathsf{T}}\mathbf{x}'$ over $10^6$ independent copies of $\mathbf{x}'$, for three different LFMMs as in Definition 1 with $n = 600$, $p = 200$, $\rho = 0.5$, $s = [\sqrt{2}; \mathbf{0}_{p-1}]$ (so that $q = 1$), and Haar distributed $\mathbf{V}$. **Left**: normal $e_1$ and uniformly distributed $e_2, \ldots, e_p$; **Middle**: normal $e_1, \ldots, e_p$; **Right**: uniformly distributed $e_1$, and normal $e_2, \ldots, e_p$.

### 5.1 GAUSSIAN UNIVERSALITY OF IN-DISTRIBUTION PERFORMANCE

Notice from (18) in Corollary 1 that the in-distribution generalization performance of $\hat{\boldsymbol{\beta}}$ under an LFMM $\mathcal{D}_{(\mathbf{x},y)}$ is determined by the random variable $r$ in (9), the distribution of which depends solely on (i) the distributions of $y, e_1, \ldots, e_q$ in the LFMM and (ii) the values of $m, \sigma^2, \psi_1, \ldots, \psi_q$ given in (13). Remark also that the values of $m, \sigma^2, \psi_1, \ldots, \psi_q$ in (13) are determined by the system of equations in (10), which concerns only the distributions of $r, y, e_1, \ldots, e_q$, as well as the deterministic parameters $\boldsymbol{\mu}, \boldsymbol{\Sigma}, \mathbf{v}_1, \ldots, \mathbf{v}_q$ of the LFMM.

We thus conclude that the distribution of $r$ is *insensitive* to the distributions of noise factors $z_{q+1}, \ldots, z_p$. In other words, an LFMM with Gaussian noises $e_1, \ldots, e_q$ in its informative factors $z_1, \ldots, z_q$ has the *same* asymptotic generalization performance as its equivalent GMM in Definition 2, *regardless of* the distributions of the noise factors $z_{q+1}, \ldots, z_p$.

A similar conclusion can be drawn from (19) of Corollary 1 on the asymptotic in-distribution training performance, by studying also the distribution of $r$ but through a proximal mapping $\text{prox}_{\kappa,\ell(\cdot,y)}$. We formalize these conclusions on the universality of in-distribution performance in Corollary 2, the proof of which is given in Appendix A.2.2.

**Corollary 2** (Condition of Gaussian universality on in-distribution performance). *Under the settings and notations of Theorem 1 and Definition 2, the Gaussian universality of in-distribution performance in Definition 3 holds if and only if noises $e_1, \ldots, e_q$ of LFMM informative factors in (5) are Gaussian.*

Figure 1 provides numerical illustrations of Corollary 2, where we compare the empirical histograms and the asymptotic distributions of the out-of-sample predicted scores $\hat{\boldsymbol{\beta}}^{\mathsf{T}}\mathbf{x}'$ for data drawn from three different LFMMs: an LFMM satisfying the in-distribution performance universality condition in Corollary 2 (**left**), an LFMM sharing the same parameters $(\boldsymbol{\mu}, \boldsymbol{\Sigma}, \rho)$ with the first but violating the condition in Corollary 2 (**right**), and their equivalent GMM in the sense of Definition 2 (**middle**).

**Remark 2** (Connection to conditional one-directional CLT in (3)). Our universality results on the in-distribution performance in Corollary 2 are related to the CGEP proven by Dandi et al. (2024) under the presumed validity of a conditional one-directional CLT in (3). Under our notations, the conditional one-directional CLT in (3) translates to the convergence of $y'\hat{\boldsymbol{\beta}}^{\mathsf{T}}\mathbf{x}'$ and $y'\hat{\boldsymbol{\beta}}_{\mathbf{g}}^{\mathsf{T}}\mathbf{x}'$ to the same normal distribution, i.e.,

$$\frac{y'\mathbf{x}'^{\mathsf{T}}\hat{\boldsymbol{\beta}} - m}{\sqrt{\sigma^2 + \sum_{k=1}^{q} \psi_k^2}} \xrightarrow{\mathrm{d}} \mathcal{N}(0,1), \qquad \frac{y'\mathbf{g}'^{\mathsf{T}}\hat{\boldsymbol{\beta}}^{\mathbf{g}} - m}{\sqrt{\sigma^2 + \sum_{k=1}^{q} \psi_k^2}} \xrightarrow{\mathrm{d}} \mathcal{N}(0,1), \tag{23}$$

as it can be shown from (14),(17) that $y'\hat{\boldsymbol{\beta}}^{\mathsf{T}}\mathbf{x}'$ has asymptotically the same distribution as $yr$, which is of mean $m$ and variance $\sigma^2 + \sum_{k=1}^{q} \psi_k^2$. It is easy to see from (9) that $yr$ is normally distributed if and only if $e_1, \ldots, e_q$ are Gaussian, which is exactly the condition of universality stated in Corollary 2.

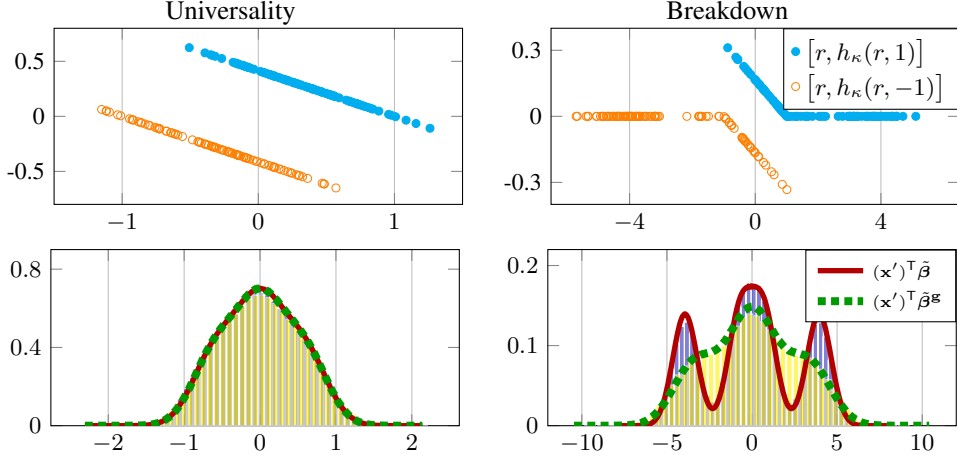

Figure 2: Empirical and theoretical results under an LFMM with $p = 200$, $\rho = 0.5$, $s = [\sqrt{2}; \mathbf{0}_{p-1}]$, Rademacher $e_1$, normal $e_2, \ldots, e_p$, and Haar distributed $\mathbf{V}$. **Top**: scatter plot of 200 independent $[r, h_\kappa(r, \pm 1)]$. **Bottom**: histograms of predicted scores on $10^6$ fresh samples $(\mathbf{x}', y') \sim \mathcal{D}_{(\mathbf{x}, y)}$ given by $\hat{\boldsymbol{\beta}}$ and $\hat{\boldsymbol{\beta}}^{\mathbf{g}}$, versus theoretical densities obtained from Theorem 1. **Left**: $n = 100$, square loss $\ell(\hat{y}, y) = (\hat{y} - y)^2/2$. **Right**: $n = 600$, square hinge loss $\ell(\hat{y}, y) = \max\{0, (1 - \hat{y}y)\}^2$.

## 5.2 GAUSSIAN UNIVERSALITY OF CLASSIFIER AND IMPLICATION FOR CHOICE OF LOSS

As discussed in Section 5.1, the system of equations in (10) does not depend on the distributions of noise factors $z_{q+1}, \ldots, z_p$. As the distribution of the high-dimensional equivalent $\tilde{\boldsymbol{\beta}}$ to $\hat{\boldsymbol{\beta}}$ given in (16) is controlled by the constants $\theta, \eta, \gamma, \omega_1, \ldots, \omega_q$ that are determined by (10), it is therefore also *universal* over the distributions of $z_{q+1}, \ldots, z_p$. Then, by a similar reasoning to Corollary 2, we conclude that the Gaussian universality of classifier in Definition 3 holds in the case of normally distributed $e_1, \ldots, e_q$.

This is however *not* the only case of Gaussian universality on classifier. Note from (9) and (10) that, even though the system of equations in (10) *does* depend on the distributions of $e_1, \ldots, e_q$, it only involves their means and variances if $h_\kappa(r, y)$ is linear in $r$. Remark also from (8) that $h_\kappa(r, y)$ varies linearly with $r$ if and only if the square loss $\ell(\hat{y}, y) = (\hat{y} - y)^2/2$ (or its rescaled version) is used.

These two conditions for the Gaussian universality of classifier as understood in Definition 3 are made precise in the following result, proven in Appendix A.2.3.

**Corollary 3** (Condition of Gaussian universality on classifier). *Under the settings and notations of Theorem 1 and Definition 2, the Gaussian universality of classifier in Definition 3 holds if and only if one of the following two conditions is met: (i) $e_1, \ldots, e_q$ in (5) are normally distributed; (ii) $\partial \ell(\hat{y}, y)/\partial \hat{y}$ is a linear function of $\hat{y}$, e.g., $\ell(\hat{y}, y) = (\hat{y} - y)^2/2$.*

**Remark 3** (Limitation of square loss). As an important consequence of Corollary 3, any classifier $\hat{\boldsymbol{\beta}}$ trained using the square loss on generic LFMM samples $\{(\mathbf{x}_i, y_i)\}_{i=1}^n \sim \mathcal{D}_{(\mathbf{x}, y)}$ and $\hat{\boldsymbol{\beta}}^{\mathbf{g}}$ trained on equivalent GMM samples $\{(\mathbf{g}_i, y_i)\}_{i=1}^n \sim \mathcal{D}_{(\mathbf{g}, y)}$ have asymptotically the same probability of correctly classifying a fresh LFMM test sample $(\mathbf{x}', y') \sim \mathcal{D}_{(\mathbf{x}, y)}$. That is, ERM classifiers trained with square loss are *unable* to adapt to non-Gaussian informative factors of LFMM, contrarily to other (non-square) losses.

The particular effect of square loss discussed in Remark 3 is numerically demonstrated in Figure 2. On the left hand side, the square loss $\ell_{\text{sqr}}(\hat{y}, y) = (\hat{y} - y)^2/2$ is used, and $h_\kappa(r, y)$ varies linearly with $r$ as in the top left plot (the two elongated scatter plots are associated respectively with $y = \pm 1$), so that the distribution of $\mathbf{x}'^\top \hat{\boldsymbol{\beta}}_{\ell_{\text{sqr}}, \lambda}$ and $\mathbf{x}'^\top \hat{\boldsymbol{\beta}}^{\mathbf{g}}_{\ell_{\text{sqr}}, \lambda}$ are *indistinguishable* in the bottom left plot of Figure 2; On the right hand, the square hinge loss $\ell_{\text{shg}}(\hat{y}, y) = \max\{0, 1 - \hat{y}y\}^2$ is used, and we observe drastically different behaviors for $\mathbf{x}'^\top \hat{\boldsymbol{\beta}}_{\ell_{\text{shg}}, \lambda}$ and $\mathbf{x}'^\top \hat{\boldsymbol{\beta}}^{\mathbf{g}}_{\ell_{\text{shg}}, \lambda}$ in the right column of Figure 2, when the points $[r, h_\kappa(r, \pm 1)]$ are highly nonlinear.

Remark 3, supported by the numerically results in Figure 2, points to the insensitivity of least-square classifiers to the distributions of non-Gaussian informative factors, despite their non-universal impact

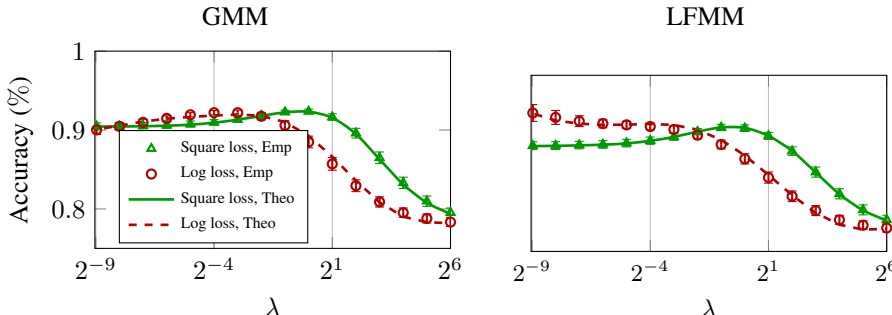

Figure 3: Empirical classification accuracy of $\hat{\boldsymbol{\beta}}_{\ell,\lambda}$ computed over $10^5$ independent copies of $(\mathbf{x}', y') \sim \mathcal{D}_{(\mathbf{x},y)}$ and averaged over 100 trials with a width of $\pm 1$ standard deviation, versus theoretical performance given in Theorem 1, given by the square loss $\ell(\hat{y}, y) = (y - \hat{y})^2/2$ and the logistic loss $\ell(\hat{y}, y) = -\ln(1/(1 + e^{-y\hat{y}}))$ and on $n = 800$ training samples. **Left**: GMM under Definition 1 with $p = 200$, $\rho = 0.5$, $s = [1, 5; 0.5; \mathbf{0}_{p-2}]$ (so that $q = 2$), and $\mathbf{V} = \mathrm{diag}(2, \mathbf{1}_{p-1})\mathbf{H}$ with Haar distributed $\mathbf{H}$. **Right**: LFMM identical to the GMM in the left, but with Rademacher $e_1$.

on the in-distribution performance as discussed in Section 5.1. The incapacity of square loss to account for non-Gaussian variations in the informative factors sheds light on the suboptimality of square loss observed in the right display of Figure 3, where the logistic loss yields better performance than the square loss with *optimally chosen* regularization $\lambda$ on LFMM having non-Gaussian informative factors, while the logistic loss *fails* to do better than the square loss under the equivalent GMM in the left-hand figure. Further experiments on real-world datasets are given in Appendix B.

This finding on the suboptimality of square loss under LFMM provide new insights on the impact of loss function beyond previous optimality results of square loss under GMM. For high-dimensional GMM data, the square loss has been proven optimal, see (Taheri et al., 2021b) for the case of unregularized ERM, and (Mai & Liao, 2019) for ridge-regularized ERM, in the $n, p \to \infty$ limit. That is, the square loss not only gives the best unbiased classifier, but also allows for an *optimal* bias-variance trade-off with well calibrated ridge-regularization. As a result of the Gaussian universality breakdown discussed above, the optimality of square loss is no longer valid under the more general LFMM. This motivates a few open questions on the optimal loss:

- Is the square loss optimal *only* under GMM, or when the Gaussian universality of in-distribution performance in Definition 3 holds?

- In the case of Gaussian universality breakdown, does the optimal loss depend on the sample ratio $n/p$ as in the setting of linear regression in (Bean et al., 2013)?

- Is it possible, in the large $n, p$ regime, to propose an optimal design of classification loss adapted to the data distribution and sample size?

## 6 CONCLUDING REMARKS

Our analysis considered a basic framework of linear factor mixture models (LFMM) and showed that the Gaussian universality can already break down under this natural extension of GMM. Based on the precise performance characterization, we derived conditions of Gaussian universality to shed light on the limit of the widely observed and extensively studied Gaussian universality phenomenon.

Breaking the Gaussian universality in classification of mixture models allows also deeper insight into the choice of classification loss beyond the optimality of square loss under GMM (Taheri et al., 2021b; Mai & Liao, 2019). The suboptimality of square loss under LFMM can be further investigated in future works, to propose, for instance, an optimal design of loss function that takes into account the data distribution and the sample size, as done by Bean et al. (2013) for linear regression.

Several simplifications made in our analysis can be removed more or less easily. For instance, while the extension to multi-classification is fairly straightforward, the generalization to non-smooth losses is less direct: even though our system of equations in (10) does not require access to the derivatives of the loss function, they are involved in the establishment of these equations.

ACKNOWLEDGMENTS

Z. Liao would like to acknowledge the National Natural Science Foundation of China (via fund NSFC-62206101 and NSFC-12141107) and the Guangdong Provincial Key Laboratory of Mathematical Foundations for Artificial Intelligence (2023B1212010001) for providing partial support. This work is also supported by the AI Cluster ANITI (ANR-19-PI3A-0004).

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

# A  PROOFS

Here, we present the detailed proofs of our theoretical results in the paper. Precisely, the proof of Theorem 1 is given in Appendix A.1 and the proofs of corollaries are given in Appendix A.2 (the proof of Corollary 1 in Appendix A.2.1, the proof of Corollary 2 in Appendix A.2.2 and that of Corollary 3 in Appendix A.2.3, respectively).

**Notations.** Before getting into the proofs, we first introduce the following asymptotic notations. The big $O$ notation $O(u_n)$ is understood here in probability. We specify that when multidimensional objects are concerned, $O(u_n)$ is understood entry-wise. The notation $O_{\|\cdot\|}(\cdot)$ is understood as follows: for a vector $\mathbf{v}$, $\mathbf{v} = O_{\|\cdot\|}(u_n)$ means its Euclidean norm is $O(u_n)$ and for a square matrix $\mathbf{M}$, $\mathbf{M} = O_{\|\cdot\|}(u_n)$ means that the operator norm of $\mathbf{M}$ is $O(u_n)$. The small $o$ notation and Big-Theta $\Theta$ are understood likewise. Note that under Assumption 3 it is equivalent to use either $O(u_n)$ or $O(u_p)$ since $n, p$ scales linearly. In the following we shall use constantly $O(u_p)$ for simplicity of exposition. The symbol $\simeq$ is used in the following sense: for a scalar $s = O(1)$, $s \simeq \tilde{s}$ indicates that $s - \tilde{s} = o(1)$, and for a vector $\mathbf{v}$ with $\|\mathbf{v}\| = O(1)$, $\mathbf{v} \simeq \tilde{\mathbf{v}}$ means $\|\mathbf{v} - \tilde{\mathbf{v}}\| = o(1)$. For random variable $r \sim \mathcal{N}(m, \sigma^2)$ with potentially random $m$ and $\sigma^2$, the expectation $\mathbb{E}[f(r)]$ should be understood as conditioned on $m, \sigma^2$, so that, $\mathbb{E}[r]$ is equal to $m$ instead of $\mathbb{E}[m]$. When parametrized functions $f_\tau(\cdot)$ are involved, $\mathbb{E}[f_\tau(r)]$ is computed by taking the integral over $r$.

## A.1  PROOF OF THEOREM 1

In this section, we will provide the proof of Theorem 1. We start by explaining the main idea of leave-one-out and laying out the key steps as a guide of our proof.

### A.1.1  MAIN IDEA AND KEY STEPS

Taking $\lambda > 0$ in the optimization problem (4) ensures a unique solution $\hat{\boldsymbol{\beta}}$. Cancelling the gradient (with respect to $\boldsymbol{\beta}$) of the objective function in (4), we obtain the following stationary-point expression of $\hat{\boldsymbol{\beta}}$

$$\lambda\hat{\boldsymbol{\beta}} = -\frac{1}{n}\sum_{i=1}^{n}\ell'(\hat{\boldsymbol{\beta}}^\mathsf{T}\mathbf{x}_i, y_i)\mathbf{x}_i, \tag{24}$$

where we denote $\ell'(t, y_i) = \frac{\partial \ell(t, y_i)}{\partial t}$.

To characterize the behavior of $\hat{\boldsymbol{\beta}}$ from (24), we need to assess the statistical behavior of $\hat{\boldsymbol{\beta}}^\mathsf{T}\mathbf{x}_i$, which is not directly tractable due to the intricate dependence between $\hat{\boldsymbol{\beta}}$ on $\mathbf{x}_i$ resulted from the implicit optimization (4). To tackle this complication, we make use of a "leave-one-out" version $\hat{\boldsymbol{\beta}}_{-i}$ of $\hat{\boldsymbol{\beta}}$ with respect to the $i$-th training sample $(\mathbf{x}_i, y)$, obtained by solving (4) with all the remaining $n - 1$ training samples $(\mathbf{x}_j, y_j)$ for $j \neq i$. Again we have

$$\lambda\hat{\boldsymbol{\beta}}_{-i} = -\frac{1}{n}\sum_{j \neq i}\ell'(\hat{\boldsymbol{\beta}}_{-i}^\mathsf{T}\mathbf{x}_j, y_j)\mathbf{x}_j. \tag{25}$$

This leave-one-out solution $\hat{\boldsymbol{\beta}}_{-i}$ has two crucial properties: (i) it is by definition *independent* of the left-out data sample $(\mathbf{x}_i, y_i)$; and (ii) it is asymptotically close to the original solution $\hat{\boldsymbol{\beta}}$ as removing one among $n$ training samples has a negligible effect as $n \to \infty$. These two properties imply that $\hat{\boldsymbol{\beta}}_{-i}$ and $\hat{\boldsymbol{\beta}}$ behave similarly on all training or test samples, *except* on $(\mathbf{x}_i, y_i)$, which is a training sample for $\hat{\boldsymbol{\beta}}$ and a new observation for $\hat{\boldsymbol{\beta}}_{-i}$.

Our proof relies on these two properties to derive a series of equations characterizing the limiting behavior of $\hat{\boldsymbol{\beta}}^\mathsf{T}\mathbf{x}_i$ and $\hat{\boldsymbol{\beta}}^\mathsf{T}\mathbf{x}'$. Below is an overview of our key steps to guide the readers through the proof.

**Key steps:**

1. Establishing the high-dimensional approximation

$$\hat{\boldsymbol{\beta}}^\mathsf{T}\mathbf{x}_i - \hat{\boldsymbol{\beta}}_{-i}^\mathsf{T}\mathbf{x}_i \simeq -\kappa\ell'(\hat{\boldsymbol{\beta}}^\mathsf{T}\mathbf{x}_i, y_i), \tag{26}$$

for some constant $\kappa$ independent of $i$.

2. Obtaining from (26) that

$$\hat{\boldsymbol{\beta}}^{\mathsf{T}}\mathbf{x}_i \simeq \mathrm{prox}_{\kappa, \ell(\cdot, y_i)}(\hat{\boldsymbol{\beta}}_{-i}^{\mathsf{T}}\mathbf{x}_i, y_i),$$

and therefore

$$-\ell'(\hat{\boldsymbol{\beta}}^{\mathsf{T}}\mathbf{x}_i, y_i) \simeq h(\hat{\boldsymbol{\beta}}_{-i}^{\mathsf{T}}\mathbf{x}_i, y_i) \equiv \frac{\mathrm{prox}_{\kappa, \ell(\cdot, y_i)}(\hat{\boldsymbol{\beta}}_{-i}^{\mathsf{T}}\mathbf{x}_i, y_i) - \hat{\boldsymbol{\beta}}_{-i}^{\mathsf{T}}\mathbf{x}_i}{\kappa},$$

where we recall $h(t, y) = (\mathrm{prox}_{\kappa, \ell(\cdot, y)}(t) - t)/\kappa$ with proximal operator $\mathrm{prox}_{\tau, f}(t) = \arg\min_{a \in \mathbb{R}} \left[ f(a) + \frac{1}{2\tau}(a - t)^2 \right]$, for $\tau > 0$ and convex $f \colon \mathbb{R} \to \mathbb{R}$.

3. Using the approximation in Step 2 to rewrite (24) as

$$\lambda\hat{\boldsymbol{\beta}} \simeq \frac{1}{n} \sum_{i=1}^{n} h(\hat{\boldsymbol{\beta}}_{-i}^{\mathsf{T}}\mathbf{x}_i, y_i)\mathbf{x}_i,$$

for

$$\mathbf{x}_i = \mathbf{V}\mathbf{z}_i = y_i\boldsymbol{\mu} + \mathbf{V}\mathbf{e}_i,$$

with $\mathbf{e}_i$ the noise vector $\mathbf{e} = [e_1, \ldots, e_p]^{\mathsf{T}} \in \mathbb{R}^p$ for $\mathbf{x}_i$, thereby replacing the intractable $\hat{\boldsymbol{\beta}}^{\mathsf{T}}\mathbf{x}_i$ in (24) with a tractable function of $\hat{\boldsymbol{\beta}}_{-i}^{\mathsf{T}}\mathbf{x}_i$.

4. Demonstrating the concentration result

$$\frac{1}{n} \sum_{i=1}^{n} h(\hat{\boldsymbol{\beta}}_{-i}^{\mathsf{T}}\mathbf{x}_i, y_i)y_i \simeq \eta,$$

for some deterministic $\eta$.

5. Demonstrating the concentration results

$$\frac{1}{n} \sum_{i=1}^{n} h(\hat{\boldsymbol{\beta}}_{-i}^{\mathsf{T}}\mathbf{x}_i, y_i)[\mathbf{e}_i]_k \simeq \phi_k, \quad \forall k \in \{1, \ldots, q\},$$

for some deterministic $\phi_1, \ldots, \phi_q$, and

$$\frac{1}{n} \sum_{i=1}^{n} h(\hat{\boldsymbol{\beta}}_{-i}^{\mathsf{T}}\mathbf{x}_i, y_i)[\mathbf{e}_i]_k \simeq 0, \quad \forall k \in \{q + 1, \ldots, p\}.$$

6. Demonstrating with concentration arguments and CLT that

$$\frac{1}{n} \sum_{i=1}^{n} h(\hat{\boldsymbol{\beta}}_{-i}^{\mathsf{T}}\mathbf{x}_i, y_i)\tilde{\mathbf{e}}_i \simeq -\theta \cdot \mathbf{V}_{\mathrm{noise}}\hat{\boldsymbol{\beta}} + \boldsymbol{\epsilon},$$

where $\tilde{\mathbf{e}}_i = [\mathbf{e}_i]_{q+1:p}$, $\mathbf{V}_{\mathrm{noise}} = [\mathbf{v}_{q+1}, \ldots, \mathbf{v}_p]$ and $\boldsymbol{\epsilon} \in \mathbb{R}^{p-q}$ a random vector such that, for any deterministic vector $\mathbf{t} = [t_{q+1}, \ldots, t_p]^{\mathsf{T}} \in \mathbb{R}^{p-q}$ of unit norm,

$$\sqrt{n}\mathbf{t}^{\mathsf{T}}\boldsymbol{\epsilon}/\gamma \to \mathcal{N}(0, 1)$$

in distribution, for some deterministic $\theta$ and $\gamma$.

7. Demonstrating

$$\kappa \simeq \frac{1}{n} \mathrm{tr}\,\boldsymbol{\Sigma}(\lambda\mathbf{I}_p + \theta\boldsymbol{\Sigma}).$$

8. Establishing asymptotic equations on $\eta, \theta, \gamma$ and $\phi$ from the results of the above steps, which characterize the limiting behavior of the solution $\hat{\boldsymbol{\beta}}$.

In the following, we present the detailed proof of Theorem 1.

### A.1.2 Detailed proof of Theorem 1

We start by establishing the following bound on the difference between $\hat{\boldsymbol{\beta}}$ and its leave-one-out version $\hat{\boldsymbol{\beta}}_{-i}$.

**Lemma 1** (Bound on $\|\hat{\boldsymbol{\beta}} - \hat{\boldsymbol{\beta}}_{-i}\|$). *For $\hat{\boldsymbol{\beta}} \in \mathbb{R}^p$ the unique solution to (4) and $\hat{\boldsymbol{\beta}}_{-i} \in \mathbb{R}^p$ the associated leave-one-out solution as defined in (25) that is independent of $\mathbf{x}_i$, we have,*

$$\left\| \hat{\boldsymbol{\beta}} - \hat{\boldsymbol{\beta}}_{-i} \right\| = O(p^{-1/2}), \tag{27}$$

*in probability as $n, p \to \infty$.*

*Proof of Lemma 1.* We define first

$$R_j = \hat{\boldsymbol{\beta}}^\mathsf{T} \mathbf{x}_j,$$
$$c_j = - \ell'(\hat{\boldsymbol{\beta}}^\mathsf{T} \mathbf{x}_j, y_j),$$

for all $j \in \{1, \ldots, n\}$, and their leave-one-out versions

$$R_{j(-i)} = \hat{\boldsymbol{\beta}}_{-i}^\mathsf{T} \mathbf{x}_j,$$
$$c_{j(-i)} = - \ell'(\hat{\boldsymbol{\beta}}_{-i}^\mathsf{T} \mathbf{x}_j, y_j),$$

for all $j \in \{1, \ldots, n\}$.

According to Assumption 1, $\ell(\cdot, y)$ is continuously differentiable and has bounded second derivative except on a finite points of set, therefore there exists universal constant $K$ such that $|\ell'(t_1) - \ell'(t_2)| \leq K|t_1 - t_2|$. As a result, for every pair of $i, j \in \{1, \ldots, n\}$, there exists a finite positive (due to the convexity of $\ell(\cdot, y)$) value $a_{j(-i)}$ such that

$$c_j - c_{j(-i)} = -a_{j(-i)} \left( \hat{\boldsymbol{\beta}}^\mathsf{T} \mathbf{x}_j - \hat{\boldsymbol{\beta}}_{-i}^\mathsf{T} \mathbf{x}_j \right). \tag{28}$$

Taking (24)$-$(25), we obtain

$$\lambda \hat{\boldsymbol{\beta}} - \lambda \hat{\boldsymbol{\beta}}_{-i} = \frac{1}{n} c_i \mathbf{x}_i + \frac{1}{n} \sum_{j \neq i} (c_j - c_{j(-i)}) \mathbf{x}_j$$

$$= \frac{1}{n} c_i \mathbf{x}_i - \left( \frac{1}{n} \sum_{j \neq i} a_{j(-i)} \mathbf{x}_j \mathbf{x}_j^\mathsf{T} \right) (\hat{\boldsymbol{\beta}} - \hat{\boldsymbol{\beta}}_{-i}).$$

Therefore

$$\left( \lambda \mathbf{I}_p + \frac{1}{n} \sum_{j \neq i} a_{j(-i)} \mathbf{x}_j \mathbf{x}_j^\mathsf{T} \right) (\hat{\boldsymbol{\beta}} - \hat{\boldsymbol{\beta}}_{-i}) = \frac{1}{n} c_i \mathbf{x}_i,$$

and

$$\hat{\boldsymbol{\beta}} - \hat{\boldsymbol{\beta}}_{-i} = \left( \lambda \mathbf{I}_p + \frac{1}{n} \sum_{j \neq i} a_{j(-i)} \mathbf{x}_j \mathbf{x}_j^\mathsf{T} \right)^{-1} \frac{1}{n} c_i \mathbf{x}_i. \tag{29}$$

Since $\frac{1}{n} \sum_{j \neq i} a_{j(-i)} \mathbf{x}_j \mathbf{x}_j^\mathsf{T}$ is non-negative definite, all eigenvalues of $\left( \lambda \mathbf{I}_p + \frac{1}{n} \sum_{j \neq i} a_{j(-i)} \mathbf{x}_j \mathbf{x}_j^\mathsf{T} \right)$ are greater than or equal to $\lambda$, so that

$$\left\| \hat{\boldsymbol{\beta}} - \hat{\boldsymbol{\beta}}_{-i} \right\| \leq \frac{1}{\lambda n} c_i \|\mathbf{x}_i\| = O(p^{-\frac{1}{2}}),$$

where we use the fact that $\hat{\boldsymbol{\beta}}$ has bounded norm as $n, p \to \infty$, which is easy to check for $\lambda > 0$, to prove the boundedness of $c_i$. This concludes the proof of Lemma 1. $\qquad\square$

As a consequence of the proof of Lemma 1, we have, by (28) that

$$a_{j(-i)} = \ell''(\hat{\boldsymbol{\beta}}_{-i}^{\mathsf{T}}\mathbf{x}_j, y) + O(p^{-\frac{1}{2}})$$

where $\ell''(t,y) = \frac{\partial \ell'(t,y)}{\partial t}$. This equation is however only valid when $\ell''(\cdot, y)$ exists, while in Assumption 1, we allow $\ell''(\cdot, y)$ to not exist on a finite set of points. Actually, as the number $l$ of $R_i$ falling on this set of point is also finite, we can take $\ell''(t, y) = \frac{\partial_- \ell'(t,y)}{\partial_- t}$ without any asymptotic impact on our results. To see that $l$ is finite, we use (24) to establish $n$ linearly independent equations on $R_1, \ldots, R_n$:

$$\lambda R_i = -\frac{1}{n}\sum_{j=1}^{n}\ell'(R_j, y_j)\mathbf{x}_i^{\mathsf{T}}\mathbf{x}_j, \quad \forall i \in \{1, \ldots, n\}.$$

As $\mathbf{x}_1, \ldots, \mathbf{x}_n$ are i.i.d. feature vectors drawn from the high-dimensional LFMM, the number of $R_i$ having the same value is finite with probability 1 at large $p$.

With a slight abuse of notation, let us set from now on

$$a_{j(-i)} = \ell''(\hat{\boldsymbol{\beta}}_{-i}^{\mathsf{T}}\mathbf{x}_j, y).$$

Then, (29) writes

$$\hat{\boldsymbol{\beta}} - \hat{\boldsymbol{\beta}}_{-i} = \frac{1}{n}c_i\mathbf{G}_{-i}^{-1}\mathbf{x}_i + O_{\|\cdot\|}(p^{-1}), \tag{30}$$

where

$$\mathbf{G}_{-i} = \lambda\mathbf{I}_p + \frac{1}{n}\sum_{j\neq i}a_{j(-i)}\mathbf{x}_j\mathbf{x}_j^{\mathsf{T}}.$$

Notice that $\mathbf{G}_{(-i)}$ is independent of $(\mathbf{x}_i, y_i)$. For $R_i = \hat{\boldsymbol{\beta}}^{\mathsf{T}}\mathbf{x}_i$ and $r_i = R_{i(-i)} = \hat{\boldsymbol{\beta}}_{-i}^{\mathsf{T}}\mathbf{x}_i$, we have, by the law of large numbers on $\mathbf{x}_i$ (recall that $\mathbb{E}[\mathbf{x}_i\mathbf{x}_i^{\mathsf{T}}] = \boldsymbol{\mu}\boldsymbol{\mu}^{\mathsf{T}} + \boldsymbol{\Sigma}$), that

$$R_i - r_i = \frac{1}{n}c_i\mathbf{x}_i^{\mathsf{T}}\mathbf{G}_{-i}^{-1}\mathbf{x}_i + O(p^{-\frac{1}{2}}) = \frac{1}{n}c_i\mathbf{e}_i^{\mathsf{T}}\mathbf{V}^{\mathsf{T}}\mathbf{G}_{-i}^{-1}\mathbf{V}\mathbf{e}_i + O(p^{-\frac{1}{2}})$$

$$= \frac{1}{n}c_i\,\mathrm{tr}\left(\mathbf{G}_{-i}^{-1}\boldsymbol{\Sigma}\right) + O(p^{-\frac{1}{2}}), \tag{31}$$

where the last equality is a classical concentration result as a consequence of the independent entries in $\mathbf{e}_i$. It is understandable that $r_i$ is significantly different from $R_i$, as the latter is the predicted score of $\hat{\boldsymbol{\beta}}$ on one of its training sample and the former the predicted score of $\hat{\boldsymbol{\beta}}_{-i}$ on a test sample independent of its training set.

Let us define

$$\kappa_i = \frac{1}{n}\,\mathrm{tr}\left(\mathbf{G}_{-i}^{-1}\boldsymbol{\Sigma}\right), \quad \kappa = \frac{1}{n}\,\mathrm{tr}\left(\mathbf{G}^{-1}\boldsymbol{\Sigma}\right),$$

where

$$\mathbf{G} = \lambda\mathbf{I}_p + \frac{1}{n}\sum_{i=1}^{n}a_i\mathbf{x}_i\mathbf{x}_i^{\mathsf{T}},$$

with $a_j = \ell''(\hat{\boldsymbol{\beta}}^{\mathsf{T}}\mathbf{x}_j, y)$. It follow from (30) that $\hat{\boldsymbol{\beta}}^{\mathsf{T}}\mathbf{x}_j - \hat{\boldsymbol{\beta}}_{-i}^{\mathsf{T}}\mathbf{x}_j = O(p^{-\frac{1}{2}})$, therefore $a_j = a_{j(-i)} + O(p^{-\frac{1}{2}})$. It is then easy to check that

$$\kappa_i = \kappa + O(p^{-\frac{1}{2}}).$$

We can thus rewrite (31) as

$$R_i - r_i = \kappa c_i + O(p^{-\frac{1}{2}}). \tag{32}$$

We arrive thus at the end of Step 1. At this point, we do not have access to the statistical behavior of $\kappa$, only the fact that it is independent of the data index $i$.

Recall

$$h_\kappa(t, y) = (\mathrm{prox}_{\kappa, \ell(\cdot, y)}(t) - t)/\kappa, \tag{33}$$

we obtain from (32)

$$c_i = h_\kappa(r_i, y_i) + O(p^{-\frac{1}{2}}).$$

It then follow from the above equation and (24) that

$$\lambda\hat{\boldsymbol{\beta}} = \frac{1}{n}\sum_{i=1}^{n} h_\kappa(r_i, y)\mathbf{x}_i + O_{\|\cdot\|}(p^{-\frac{1}{2}}). \tag{34}$$

Set from now on

$$c_i = h_\kappa(r_i, y_i),$$

and rewrite (24) as

$$\lambda\hat{\boldsymbol{\beta}} = \frac{1}{n}\sum_{i=1}^{n} c_i\mathbf{x}_i + O_{\|\cdot\|}(p^{-\frac{1}{2}}). \tag{35}$$

We arrive thus at the end of Step 3.

We will now demonstrate

$$\frac{1}{n}\sum_{i=1}^{n} y_i c_i = \mathbb{E}[y_i c_i] + O(p^{-\frac{1}{4}}) \tag{36}$$

by showing the variance of $\frac{1}{n}\sum_{i=1}^{n} y_i c_i$ is of order $O(p^{-\frac{1}{2}})$.

To do so, we need to introduce the definition of leave-two-out solution $\hat{\boldsymbol{\beta}}_{-ij}$ obtained by removing not one but two training samples $(\mathbf{x}_i, y_i)$ and $(\mathbf{x}_j, y_j)$. The subscript $-ij$ is understood similarly to the subscript $-i$, but associated with the statistical objects dependent of $\hat{\boldsymbol{\beta}}_{-ij}$.

Similarly to (30), we have

$$\hat{\boldsymbol{\beta}}_{-i} - \hat{\boldsymbol{\beta}}_{-ij} = \frac{1}{n}c_{(-i)j}\mathbf{G}_{-ij}^{-1}\mathbf{x}_j + O_{\|\cdot\|}(p^{-1}), \tag{37}$$

where

$$c_{(-i)j} = h_\kappa(r_{(-i)j}, y), \quad r_{(-i)j} = \hat{\boldsymbol{\beta}}_{-ij}^{\mathsf{T}}\mathbf{x}_j.$$

Multiplying (37) with $\mathbf{x}_i^{\mathsf{T}}$ from the left side, we get

$$r_i - r_{(-j)i} = \frac{1}{n}c_{(-i)j}\mathbf{x}_i^{\mathsf{T}}\mathbf{G}_{-ij}^{-1}\mathbf{x}_j + O(p^{-\frac{1}{2}}) = O(p^{-\frac{1}{2}}).$$

Then for $i \neq j$, we observe from the above equation that

$$\mathbb{E}[y_i c_i y_j c_j] = \mathbb{E}[y_i y_j h_\kappa(r_i, y_i) h_\kappa(r_j, y_j)]$$
$$= \mathbb{E}[y_i y_j h_\kappa(r_{(-j)i}, y_i) h_\kappa(r_{(-i)j}, y_j)] + O(p^{-\frac{1}{2}}).$$

Note importantly that, conditioned on $\hat{\boldsymbol{\beta}}_{-ij}$, $r_{(-j)i}$ and $r_{(-i)j}$ are independent. We have thus

$$\mathbb{E}[y_i y_j h_\kappa(r_{(-j)i}, y_i) h_\kappa(r_{(-i)j}, y_j)] = \mathbb{E}\left[\mathbb{E}[y_i y_j h_\kappa(r_{(-j)i}, y_i) h_\kappa(r_{(-i)j}, y_j)|\hat{\boldsymbol{\beta}}_{-ij}]\right]$$
$$= \mathbb{E}\left[\mathbb{E}[y_i h_\kappa(r_{(-j)i}, y_i)|\hat{\boldsymbol{\beta}}_{-ij}]\mathbb{E}[y_j h_\kappa(r_{(-i)j}, y_j)|\hat{\boldsymbol{\beta}}_{-ij}]\right]$$
$$= \mathbb{E}[y_i h_\kappa(r_{(-j)i}, y_i)]\mathbb{E}[y_j h_\kappa(r_{(-i)j}, y_j)].$$

Since $h_\kappa(r_{(-i)j}, y_j) = h_\kappa(r_j, y_j) + O(p^{-\frac{1}{2}})$, we get from the above equation and the one before that

$$\mathbb{E}[y_i c_i y_j c_j] = \mathbb{E}[y_i c_i]\mathbb{E}[y_j c_j] + O(p^{-\frac{1}{2}}).$$

It follow directly that

$$\mathrm{Var}\left[\frac{1}{n}\sum_{i=1}^{n} y_i c_i\right] = O(p^{-\frac{1}{2}}). \tag{38}$$

Therefore

$$\frac{1}{n}\sum_{i=1}^{n} y_i c_i = \eta + O(p^{-\frac{1}{4}}), \quad \text{with} \quad \eta \equiv \mathbb{E}[y_i c_i]. \tag{39}$$

We prove thus (39), which brings us to the end of Step 4.

By the same reasoning, we obtain also

$$\frac{1}{n}\sum_{i=1}^{n}c_i[\mathbf{e}_i]_k = \phi_k + O(p^{-\frac{1}{4}}), \quad \text{with} \quad \phi_k \equiv \mathbb{E}[c_i[\mathbf{e}_i]_k]. \tag{40}$$

To see that

$$\phi_k \simeq 0, \forall k \in \{q+1,\ldots,p\},$$

we define first the leave-one-variable-out classifier $\hat{\boldsymbol{\beta}}^{-k}$ as the solution to (4) on a training set generated from a slightly differently distribution than $\mathcal{D}_{(\mathbf{x},y)}$, with $e_k$ constantly equal to zero. The superscript $-k$ is understood similarly to the subscript $-i$ in the statistical objects dependent of $\hat{\boldsymbol{\beta}}_{-i}$, with their leave-one-variable-out version obtained by replacing $\hat{\boldsymbol{\beta}}_{-i}$ with $\hat{\boldsymbol{\beta}}^{-k}$. Similarly to the leave-one-out reasoning with respect to the data samples, the same asymptotic arguments can be applied to control the difference between $\hat{\boldsymbol{\beta}}$ and $\hat{\boldsymbol{\beta}}^{-k}$. In the same spirit as (30), we have

$$\hat{\boldsymbol{\beta}} - \hat{\boldsymbol{\beta}}^{-k} = \frac{1}{n}\mathbf{G}^{-k}\mathbf{e}_{[k]} + O_{\|\cdot\|}(p^{-1}) \tag{41}$$

where $\mathbf{e}_{[k]} \in \mathbb{R}^n$ is a vector with its $i$-th element being $[\mathbf{e}_i]_k$, and $\mathbf{G}^{-k}$ a matrix of bounded norm with high probability and independent of $[\mathbf{e}_i]_k$.

Note first from (35) that

$$\lambda\mathbf{V}_{\text{noise}}\hat{\boldsymbol{\beta}} = \frac{1}{n}\sum_{i=1}^{n}c_i\mathbf{V}_{\text{noise}}\mathbf{x}_i + O_{\|\cdot\|}(p^{-\frac{1}{2}})$$

$$= \frac{1}{n}\sum_{i=1}^{n}c_i\mathbf{V}_{\text{noise}}\mathbf{V}_{\text{noise}}^{\mathsf{T}}\tilde{\mathbf{e}}_i + O_{\|\cdot\|}(p^{-\frac{1}{2}}) \tag{42}$$

where we have $\mathbf{V}_{\text{noise}}\mathbf{x}_i = \mathbf{V}_{\text{noise}}\mathbf{V}_{\text{noise}}^{\mathsf{T}}\tilde{\mathbf{e}}_i$ according to the orthogonality between the signal subspace and the noise subspace stated in Item (ii) of Assumption 2. As the eigenvalues of $\mathbf{V}_{\text{noise}}\mathbf{V}_{\text{noise}}^{\mathsf{T}}$ are comparable to 1 according to Item (ii) of Assumption 3, we get

$$\lambda\left(\mathbf{V}_{\text{noise}}\mathbf{V}_{\text{noise}}^{\mathsf{T}}\right)^{-1}\mathbf{V}_{\text{noise}}\hat{\boldsymbol{\beta}} = \frac{1}{n}\sum_{i=1}^{n}c_i\tilde{\mathbf{e}}_i + O_{\|\cdot\|}(p^{-\frac{1}{2}}). \tag{43}$$

Similarly, we have, for $\hat{\boldsymbol{\beta}}^{-k}$, that

$$\lambda\left(\mathbf{V}_{\text{noise}}\mathbf{V}_{\text{noise}}^{\mathsf{T}}\right)^{-1}\mathbf{V}_{\text{noise}}\hat{\boldsymbol{\beta}}^{-k} = \frac{1}{n}\sum_{i=1}^{n}c_i\tilde{\mathbf{e}}_i^{-k} + O_{\|\cdot\|}(p^{-\frac{1}{2}}). \tag{44}$$

Combining (41), (43) and (44), we obtain that, for $k \in \{q+1,\ldots,p\}$,

$$\phi_k = \lambda\left[\left(\mathbf{V}_{\text{noise}}\mathbf{V}_{\text{noise}}^{\mathsf{T}}\right)^{-1}\mathbf{V}_{\text{noise}}\mathbb{E}[\hat{\boldsymbol{\beta}}]\right]_{k-q} + O(p^{-\frac{1}{2}})$$

$$= \lambda\left[\left(\mathbf{V}_{\text{noise}}\mathbf{V}_{\text{noise}}^{\mathsf{T}}\right)^{-1}\mathbf{V}_{\text{noise}}\left(\mathbb{E}[\hat{\boldsymbol{\beta}}^{-k}] - \frac{1}{n}\mathbb{E}[\mathbf{G}^{-k}\mathbf{e}_{[k]}]\right)\right]_{k-q} + O(p^{-\frac{1}{2}})$$

$$= O(p^{-\frac{1}{2}}), \tag{45}$$

where we used the fact that

$$\left[\left(\mathbf{V}_{\text{noise}}\mathbf{V}_{\text{noise}}^{\mathsf{T}}\right)^{-1}\mathbf{V}_{\text{noise}}\mathbb{E}[\hat{\boldsymbol{\beta}}^{-k}]\right]_{k-q} = \frac{1}{n}\sum_{i=1}^{n}\mathbb{E}[c_i[\tilde{\mathbf{e}}_i^{-k}]_{k-q}] = 0,$$

since $[\tilde{\mathbf{e}}_i^{-k}]_{k-q} = [\mathbf{e}_i^{-k}]_k = 0$ for all $i \in \{1,\ldots,n\}$ according to the definition of the leave-one-variable-out classifier $\hat{\boldsymbol{\beta}}^{-k}$.

We arrive thus at the end of Step 5.

Recall from (45) that $\phi_k = O(p^{-\frac{1}{2}})$ for $k \in \{q+1, \ldots, p\}$, we have thus from (40) that

$$\frac{1}{n}\sum_{i=1}^{n} c_i[\mathbf{e}_i]_k = O(p^{-\frac{1}{4}}), \quad \forall k \in \{q+1, \ldots, p\}.$$

It follows then from (42) that

$$\hat{\boldsymbol{\beta}}^\mathsf{T}\mathbf{v}_k = O(p^{-\frac{1}{4}}), \quad \forall k \in \{q+1, \ldots, p\}. \tag{46}$$

We observe then, for $k \in \{q+1, \ldots, p\}$,

$$\frac{1}{n}\sum_{i=1}^{n} c_i[\mathbf{e}_i]_k = \frac{1}{n}\sum_{i=1}^{n} h_\kappa(r_i, y_i)[\mathbf{e}_i]_k = \frac{1}{n}\sum_{i=1}^{n} h_\kappa\left(\sum_{k=1}^{p}(\hat{\boldsymbol{\beta}}^\mathsf{T}\mathbf{v}_k)[\mathbf{e}_i]_k, y_i\right)[\mathbf{e}_i]_k$$

$$= \frac{1}{n}\sum_{i=1}^{n} h_\kappa\left(\sum_{d\neq k}(\hat{\boldsymbol{\beta}}^\mathsf{T}\mathbf{v}_d)[\mathbf{e}_i]_d, y_i\right)[\mathbf{e}_i]_k$$

$$+ \frac{1}{n}\sum_{i=1}^{n} h'_\kappa\left(\sum_{d\neq k}(\hat{\boldsymbol{\beta}}^\mathsf{T}\mathbf{v}_d)[\mathbf{e}_i]_d, y_i\right)(\hat{\boldsymbol{\beta}}^\mathsf{T}\mathbf{v}_k)[\mathbf{e}_i]_k^2 \tag{47}$$

$$+ \frac{1}{n}\sum_{i=1}^{n} h''_\kappa\left(\sum_{d\neq k}(\hat{\boldsymbol{\beta}}^\mathsf{T}\mathbf{v}_d)[\mathbf{e}_i]_d, y_i\right)(\hat{\boldsymbol{\beta}}^\mathsf{T}\mathbf{v}_k)^2[\mathbf{e}_i]_k^3 + O(p^{-\frac{3}{4}}), \tag{48}$$

where we denote $h''_\kappa(r, y) = \frac{\partial h'_\kappa(r,y)}{\partial r}$.

We denote the first term by

$$\epsilon_k = \frac{1}{n}\sum_{i=1}^{n} h_\kappa\left(\sum_{d\neq k}(\hat{\boldsymbol{\beta}}^\mathsf{T}\mathbf{v}_d)[\mathbf{e}_i]_d, y_i\right)[\mathbf{e}_i]_k.$$

Note from (41) that

$$\epsilon_k = \frac{1}{n}\sum_{i=1}^{n} h_\kappa\left(\sum_{d\neq k}(\mathbf{v}_d^\mathsf{T}\hat{\boldsymbol{\beta}}^{-k})[\mathbf{e}_i]_d, y_i\right)[\mathbf{e}_i]_k + O(p^{-1}). \tag{49}$$

Since $\hat{\boldsymbol{\beta}}^{-k}$ is by definition independent of all $[\mathbf{e}_i]_k$ for $i \in \{1, \ldots, n\}$, we notice that all $h_\kappa\left(\sum_{d\neq k}(\mathbf{v}_d^\mathsf{T}\hat{\boldsymbol{\beta}}^{-k})[\mathbf{e}_i]_d, y_i\right)[\mathbf{e}_i]_k, i \in \{1, \ldots, n\}$ are independent conditioned on $\{\mathbf{e}^{[d]}\}_{d\in\{q+1,\ldots,p\}\setminus k}$. We assess the conditional mean of $\epsilon_k$ as follows

$$\mathbb{E}\left[\epsilon_k|\{\mathbf{e}^{[d]}\}_{d\in\{q+1,\ldots,p\}\setminus k}\right] = \frac{1}{n}\sum_{i=1}^{n} h_\kappa\left(\sum_{d\neq k}(\mathbf{v}_d^\mathsf{T}\hat{\boldsymbol{\beta}}^{-k})[\mathbf{e}_i]_d, y_i\right)\mathbb{E}\left[[\mathbf{e}_i]_k\right] + O(p^{-\frac{1}{2}})$$

$$= O(p^{-1}). \tag{50}$$

Similarly we have

$$\mathrm{Var}\left[\epsilon_k|\{\mathbf{e}^{[d]}\}_{d\in\{q+1,\ldots,p\}\setminus k}\right] = \frac{1}{n^2}\sum_{i=1}^{n} h_\kappa\left(\sum_{d\neq k}(\mathbf{v}_d^\mathsf{T}\hat{\boldsymbol{\beta}}^{-k})[\mathbf{e}_i]_d, y_i\right)^2\mathbb{E}\left[[\mathbf{e}_i]_k^2\right]$$

$$= \frac{1}{n}\mathbb{E}\left[h_\kappa\left(\sum_{d\neq k}(\mathbf{v}_d^\mathsf{T}\hat{\boldsymbol{\beta}}^{-k}) + O(p^{-\frac{5}{4}})[\mathbf{e}_i]_d, y_i\right)^2\right] + O(p^{-\frac{5}{4}}),$$

where the second line is obtained by a similar reasoning to (39) We remark thus that the concentrations of the conditional mean $\mathbb{E}\left[\epsilon_k | \{ \mathbf{e}^{[d]} \}_{d \in \{q+1,\ldots,p\} \setminus k}\right]$ and variance $\mathrm{Var}\left[\epsilon_k | \{ \mathbf{e}^{[d]} \}_{d \in \{q+1,\ldots,p\} \setminus k}\right]$ around the same limits:

$$\mathbb{E}\left[\epsilon_k | \{ \mathbf{e}^{[d]} \}_{d \in \{q+1,\ldots,p\} \setminus k}\right] = O(p^{-1})$$

$$\mathrm{Var}\left[\epsilon_k | \{ \mathbf{e}^{[d]} \}_{d \in \{q+1,\ldots,p\} \setminus k}\right] = \frac{1}{n}\gamma^2 + O(p^{-\frac{5}{4}}), \quad \text{with} \quad \gamma^2 \equiv \mathbb{E}[c_i^2]. \tag{51}$$

In summary, when conditioned on $\{\mathbf{e}^{[d]}\}_{d \in \{q+1,\ldots,p\} \setminus k}$, the sum of independent $\frac{1}{n}h_\kappa\left(\sum_{d \neq k}(\mathbf{v}_d^\mathsf{T}\hat{\boldsymbol{\beta}}^{-k})[\mathbf{e}_i]_d, y_i\right)[\mathbf{e}_i]_k, i \in \{1,\ldots,n\}$ is of mean asymptotically equal to $0$ and variance asymptotically equal to $\gamma^2$. Then, by the central limit theorem, we have

$$\sqrt{n}\epsilon_k/\gamma \xrightarrow{\mathrm{d}} \mathcal{N}(0,1), \quad \forall k \in \{q+1,\ldots,p\}, \tag{52}$$

in distribution as $n, p \to \infty$ at the same pace.

Now we wish to show that

$$\sqrt{n}\sum_{k=q+1}^{p} t_k\epsilon_k/\gamma \xrightarrow{\mathrm{d}} \mathcal{N}(0,1) \tag{53}$$

for any deterministic vector $\mathbf{t} = [t_{q+1},\ldots,t_p]^\mathsf{T} \in \mathbb{R}^{p-q}$ of unit norm. To this end, we introduce the leave-two-variables-out solution $\hat{\boldsymbol{\beta}}^{-kd}$ obtained similarly to $\hat{\boldsymbol{\beta}}^{-k}$ but with both $e_k, e_d$ constantly set to $0$. The superscript $-kd$ is understood similarly to the superscript $-k$, but associated with the statistical objects dependent of $\hat{\boldsymbol{\beta}}_{-kd}$. It is easy to see that

$$\mathbb{E}\left[\sqrt{n}\sum_{k=q+1}^{p} t_k\epsilon_k\right] = \sum_{k=q+1}^{p} t_k\mathbb{E}\left[\epsilon_k\right] = O(p^{-\frac{1}{2}}).$$

To approximate the variance of $\sqrt{n}\sum_{k=q+1}^{p} t_k\epsilon_k$, let us define first

$$\epsilon_{k_1}^{-k_2} = \frac{1}{n}\sum_{i=1}^{n}h_\kappa\left(\sum_{d \neq k_1,k_2}(\mathbf{v}_d^\mathsf{T}\hat{\boldsymbol{\beta}}^{-k_2})[\mathbf{e}_i]_d, y_i\right)[\mathbf{e}_i]_{k_1}, \tag{54}$$

for which we have

$$\epsilon_{k_1} = \epsilon_{k_1}^{-k_2} + (p^{-\frac{3}{4}}) \tag{55}$$

from (41) and (46). Similarly to (49), we have

$$\epsilon_{k_1}^{-k_2} = \frac{1}{n}\sum_{i=1}^{n}h_\kappa\left(\sum_{d \neq k_1,k_2}(\mathbf{v}_d^\mathsf{T}\hat{\boldsymbol{\beta}}^{-k_1 k_2})[\mathbf{e}_i]_d, y_i\right)[\mathbf{e}_i]_{k_1} + O(p^{-1}).$$

Therefore

$$\epsilon_{k_1} = \frac{1}{n}\sum_{i=1}^{n}h_\kappa\left(\sum_{d \neq k_1,k_2}(\mathbf{v}_d^\mathsf{T}\hat{\boldsymbol{\beta}}^{-k_1 k_2})[\mathbf{e}_i]_d, y_i\right)[\mathbf{e}_i]_{k_1} + (p^{-\frac{3}{4}})$$

$$\epsilon_{k_2} = \frac{1}{n}\sum_{i=1}^{n}h_\kappa\left(\sum_{d \neq k_1,k_2}(\mathbf{v}_d^\mathsf{T}\hat{\boldsymbol{\beta}}^{-k_1 k_2})[\mathbf{e}_i]_d, y_i\right)[\mathbf{e}_i]_{k_2} + (p^{-\frac{3}{4}}),$$

where we notice that $\sum_{i=1}^{n}h_\kappa\left(\sum_{d \neq k_1,k_2}(\mathbf{v}_d^\mathsf{T}\hat{\boldsymbol{\beta}}^{-k_1 k_2})[\mathbf{e}_i]_d, y_i\right)[\mathbf{e}_i]_{k_1}$ is independent of $\sum_{i=1}^{n}h_\kappa\left(\sum_{d \neq k_1,k_2}(\mathbf{v}_d^\mathsf{T}\hat{\boldsymbol{\beta}}^{-k_1 k_2})[\mathbf{e}_i]_d, y_i\right)[\mathbf{e}_i]_{k_2}$. We obtain thus

$$\mathrm{Var}\left[\sqrt{n}\sum_{k=q+1}^{p} t_k\epsilon_k\right] = n\sum_{k_1,k_2=q+1}^{p} t_k^2\mathbb{E}\left[\epsilon_{k_1}\epsilon_{k_2}\right] = n\sum_{k=q+1}^{p} t_k^2\mathbb{E}\left[\epsilon_k^2\right] + n\sum_{k_1 \neq k_2=q+1}^{p} t_k^2\mathbb{E}\left[\epsilon_{k_1}\epsilon_{k_2}\right]$$

$$= \gamma^2 + O(p^{-\frac{1}{4}}).$$

To obtain (53), it suffices now to demonstrate the asymptotic mutual independence of $\epsilon_{q+1}, \ldots, \epsilon_q$ by showing that $\epsilon_k$ is asymptotically independent of $\{\epsilon_d\}_{d \neq k = q+1}$ for all $k \in \{q+1, \ldots, p\}$. Let $k = q + 1$ without the loss of generality, observe from (49) and (52) that

$$\epsilon_{q+1} \simeq \frac{1}{n} \sum_{i=1}^{n} h_\kappa \left( \sum_{d \neq q+1}^{p} (\mathbf{v}_d^\mathsf{T} \hat{\boldsymbol{\beta}}^{-(q+1)}) [\mathbf{e}_i]_d, y_i \right) [\mathbf{e}_i]_{q+1},$$

and recall from (55) that

$$\begin{bmatrix} \epsilon_{q+2} \\ \epsilon_{q+3} \\ \vdots \\ \epsilon_p \end{bmatrix} \simeq \begin{bmatrix} \epsilon_{q+2}^{-(q+1)} \\ \epsilon_{q+3}^{-(q+1)} \\ \vdots \\ \epsilon_p^{-(q+1)} \end{bmatrix}.$$

It is easy to see from (54) that $\epsilon_{q+2}^{-(q+1)}, \ldots, \epsilon_p^{-(q+1)}$ is independent of $\mathbf{e}_{q+1}$. Therefore $\frac{1}{n} \sum_{i=1}^{n} h_\kappa \left( \sum_{d \neq q+1}^{p} (\mathbf{v}_d^\mathsf{T} \hat{\boldsymbol{\beta}}^{-(q+1)}) [\mathbf{e}_i]_d, y_i \right) [\mathbf{e}_i]_{q+1}$ is independent of $\epsilon_{q+2}^{-(q+1)}, \ldots, \epsilon_p^{-(q+1)}$. We obtain thus (53).

Now we turn to the second term of (48). In a similar manner to (39), we have

$$\frac{1}{n} \sum_{i=1}^{n} h_\kappa' \left( \sum_{d \neq k} (\hat{\boldsymbol{\beta}}^\mathsf{T} \mathbf{v}_d) [\mathbf{e}_i]_d, y_i \right) [\mathbf{e}_i]_k^2 = \frac{1}{n} \sum_{i=1}^{n} h_\kappa' \left( \sum_{d \neq k} (\mathbf{v}_d^\mathsf{T} \hat{\boldsymbol{\beta}}^{-k}) [\mathbf{e}_i]_d, y_i \right) [\mathbf{e}_i]_k^2 + O(p^{-1})$$

$$= \mathbb{E} \left[ h_\kappa' \left( \sum_{d \neq k} (\mathbf{v}_d^\mathsf{T} \hat{\boldsymbol{\beta}}^{-k}) [\mathbf{e}_i]_d, y_i \right) \right] \mathbb{E} \left[ [\mathbf{e}_i]_k^2 \right] + O(p^{-\frac{1}{4}}).$$

Consequently,

$$\frac{1}{n} \sum_{i=1}^{n} h_\kappa' \left( \sum_{d \neq k} (\hat{\boldsymbol{\beta}}^\mathsf{T} \mathbf{v}_d) [\mathbf{e}_i]_d, y_i \right) [\mathbf{e}_i]_k^2 \hat{\boldsymbol{\beta}}^\mathsf{T} \mathbf{v}_k = \left( -\theta + O(p^{-\frac{1}{4}}) \right) \hat{\boldsymbol{\beta}}^\mathsf{T} \mathbf{v}_k, \text{ with } \theta \equiv -\mathbb{E}[h_\kappa'(r_i, y_i)]. \tag{56}$$

To control the third term of (48), it suffices to prove that its second moment is of $o(p^{-\frac{1}{2}})$ by using the concentration arguments with the leave-one-variable-out manipulation as before, and the fact that $\hat{\boldsymbol{\beta}}^\mathsf{T} \mathbf{v}_k = O(p^{-\frac{1}{4}})$ for $k \in \{q+1, \ldots, p\}$.

We arrive thus at the end of Step 6.

Rewrite now (35) as

$$\lambda \hat{\boldsymbol{\beta}} = \frac{1}{n} \sum_{i=1}^{n} c_i y_i \boldsymbol{\mu} + \frac{1}{n} \sum_{i=1}^{n} c_i \mathbf{V} \mathbf{e}_i + O_{\|\cdot\|}(p^{-\frac{1}{2}}). \tag{57}$$

Summarizing (39), (40), (48), (52), nd (56), we obtain

$$\lambda \hat{\boldsymbol{\beta}} \simeq \eta \boldsymbol{\mu} + \sum_{k=1}^{q} \phi_k \mathbf{v}_k + \theta \mathbf{V}_{\text{noise}} \mathbf{V}_{\text{noise}}^\mathsf{T} \hat{\boldsymbol{\beta}} + \mathbf{V}_{\text{noise}} \boldsymbol{\epsilon},$$

with $\boldsymbol{\epsilon} = [\epsilon_{q+1}, \ldots, \epsilon_p]^\mathsf{T}$. Therefore

$$\hat{\boldsymbol{\beta}} \simeq (\lambda \mathbf{I}_p + \theta \boldsymbol{\Sigma})^{-1} \left( \eta \boldsymbol{\mu} + \sum_{k=1}^{q} \omega_k \mathbf{v}_k + \mathbf{V}_{\text{noise}} \boldsymbol{\epsilon} \right), \tag{58}$$

where $\omega_k \equiv \phi_k + \theta \mathbb{E}[\hat{\boldsymbol{\beta}}]^\mathsf{T} \mathbf{v}_k$, and

$$\sqrt{n} \mathbf{t}^\mathsf{T} \boldsymbol{\epsilon} / \gamma \xrightarrow{\mathrm{d}} \mathcal{N}(0, 1)$$

for any deterministic vector $\mathbf{t} = [t_{q+1}, \ldots, t_p]^\mathsf{T} \in \mathbb{R}^{p-q}$ of unit norm according to (53). Similarly, we have, for the leave-one-out solution, that

$$\hat{\boldsymbol{\beta}}_{-i} \simeq (\lambda \mathbf{I}_p + \theta \boldsymbol{\Sigma})^{-1} \left( \eta \boldsymbol{\mu} + \sum_{k=1}^{q} \omega_k \mathbf{v}_k + \mathbf{V}_{\text{noise}} \boldsymbol{\epsilon}_{-i} \right) \qquad (59)$$

where $\boldsymbol{\epsilon}_{-i} = [\epsilon_{q+1(-i)}, \ldots, \epsilon_{p(-i)}]^\mathsf{T}$ with

$$\epsilon_{k(-i)} = \frac{1}{n} \sum_{j \neq i} h_\kappa \left( \sum_{d \neq k} (\hat{\boldsymbol{\beta}}_{-i}^\mathsf{T} \mathbf{v}_d)[\mathbf{e}_j]_d, y_j \right) [\mathbf{e}_j]_k.$$

We get from (30) that

$$\epsilon_k - \epsilon_{k(-i)} \simeq \frac{1}{n} h_\kappa \left( \sum_{d \neq k} (\hat{\boldsymbol{\beta}}^\mathsf{T} \mathbf{v}_d)[\mathbf{e}_i]_d, y_i \right) [\mathbf{e}_i]_k.$$

Hence

$$\hat{\boldsymbol{\beta}} - \hat{\boldsymbol{\beta}}_{-i} \simeq (\lambda \mathbf{I}_p + \theta \boldsymbol{\Sigma})^{-1} \frac{1}{n} \mathbf{V}_{\text{noise}} c_i \tilde{\mathbf{e}}_i,$$

leading to

$$\mathbf{x}_i^\mathsf{T} \hat{\boldsymbol{\beta}} - \mathbf{x}_i^\mathsf{T} \hat{\boldsymbol{\beta}}_{-i} \simeq (y_i \boldsymbol{\mu} + \mathbf{V} \mathbf{e}_i)^\mathsf{T} (\lambda \mathbf{I}_p + \theta \boldsymbol{\Sigma})^{-1} \frac{1}{n} \mathbf{V}_{\text{noise}} c_i \tilde{\mathbf{e}}_i$$

$$\simeq \frac{c_i}{n} \operatorname{tr} \boldsymbol{\Sigma} (\lambda \mathbf{I}_p + \theta \boldsymbol{\Sigma}).$$

Comparing the above equation with (32), we observe

$$\kappa \simeq \frac{1}{n} \operatorname{tr} \boldsymbol{\Sigma} (\lambda \mathbf{I}_p + \theta \boldsymbol{\Sigma}).$$

We are now at the end of Step 7.

It is easy to see from (59) that

$$r_i = \hat{\boldsymbol{\beta}}_{-i}^\mathsf{T} \mathbf{x}_i \simeq y_i m + \sum_{k=1}^{p} \psi_k [\mathbf{e}_i]_k + \sigma \tilde{e},$$

where $\tilde{e}$ a random variable independent of $[\mathbf{e}_i]_1, \ldots, [\mathbf{e}_i]_q$ with

$$\tilde{e} \xrightarrow{\mathrm{d}} \mathcal{N}(0, 1),$$

and

$$m = \boldsymbol{\mu}^\mathsf{T} \mathbf{Q} \boldsymbol{\xi}, \quad \sigma^2 = \frac{\gamma^2}{p} \operatorname{tr} (\mathbf{Q} \boldsymbol{\Sigma})^2, \quad \psi_k = \mathbf{v}_k^\mathsf{T} \mathbf{Q} \boldsymbol{\xi}, \quad \forall k \in \{1, \ldots, q\},$$

with

$$\mathbf{Q} = (\lambda \mathbf{I}_p + \theta \boldsymbol{\Sigma})^{-1}, \quad \boldsymbol{\xi} = \eta \boldsymbol{\mu} + \sum_{k=1}^{q} \omega_k \mathbf{v}_k.$$

We obtain thus the system of equations in (10) from (39), (40), (51) and (56), which gives access to the values of $\theta, \eta, \gamma, \omega_1, \ldots, \omega_q$.

Set

$$\tilde{\boldsymbol{\beta}} \equiv (\lambda \mathbf{I}_p + \theta \boldsymbol{\Sigma})^{-1} \left( \eta \boldsymbol{\mu} + \sum_{k=1}^{q} \omega_k \mathbf{v}_k + \gamma \boldsymbol{\Sigma}^{\frac{1}{2}} \mathbf{u} \right) \qquad (60)$$

where $\mathbf{u} \sim \mathcal{N}(\mathbf{0}_p, \mathbf{I}_p/n)$. We obtain (14) from (60) and (58) by a simple application of CLT, and (15) from (60), (59) and (32).

This concludes the proof of Theorem 1.

## A.2 PROOFS OF COROLLARIES

### A.2.1 PROOF OF COROLLARY 1

Recall from Theorem 1 that

$$\tilde{\boldsymbol{\beta}} = (\lambda\mathbf{I}_p + \theta\boldsymbol{\Sigma})^{-1}\left(\eta\boldsymbol{\mu} + \sum_{k=1}^{q}\omega_k\mathbf{v}_k + \gamma\boldsymbol{\Sigma}^{\frac{1}{2}}\mathbf{u}\right),$$

for Gaussian vector $\mathbf{u} \sim \mathcal{N}(\mathbf{0}_p, \mathbf{I}_p/n)$ independent of $\{(\mathbf{x}_i, y_i)\}_{i=1}^{n}$. For $(\mathbf{x}, y) \sim \mathcal{D}_{(\mathbf{x},y)}$ independent of $\tilde{\boldsymbol{\beta}}$, we recall from Definition 1 that

$$\mathbf{x} = y\boldsymbol{\mu} + \mathbf{V}\mathbf{e},$$

with $\mathbf{e} = [e_1, \ldots, e_p]^{\mathsf{T}}$. Then, let $\mathbf{V}_{\text{noise}} = [\mathbf{v}_{q+1}, \ldots, \mathbf{v}_p]$ and $\tilde{\mathbf{e}} = [e_{q+1}, \ldots, e_p]^{\mathsf{T}}$, we have

$$\tilde{\boldsymbol{\beta}}^{\mathsf{T}}\mathbf{x} = ym + \sum_{k=1}^{q}\psi_k e_k + \boldsymbol{\xi}^{\mathsf{T}}\mathbf{Q}\mathbf{V}_{\text{noise}}\tilde{\mathbf{e}} + \gamma\mathbf{u}^{\mathsf{T}}\boldsymbol{\Sigma}^{\frac{1}{2}}\mathbf{Q}\mathbf{V}_{\text{noise}}\tilde{\mathbf{e}},$$

with $m, \psi_1, \ldots, \psi_q$ as given in (13), and $\mathbf{Q}, \boldsymbol{\xi}$ as in (11).

Note importantly that

$$\boldsymbol{\xi}^{\mathsf{T}}\mathbf{Q}\mathbf{v}_k = 0, \forall k \in \{q+1, \ldots, p\}$$

due to the orthogonality of $\text{Span}\{\mathbf{v}_1, \ldots, \mathbf{v}_q\}$ to $\text{Span}\{\mathbf{v}_{q+1}, \ldots, \mathbf{v}_p\}$ stated in Item(ii) of Assumption 2

As $\mathbf{u}, \tilde{\mathbf{e}}$ are independent random vectors of independent entries, we have, by CLT, that

$$\frac{\gamma\mathbf{u}^{\mathsf{T}}\boldsymbol{\Sigma}^{\frac{1}{2}}\mathbf{Q}\mathbf{V}_{\text{noise}}\tilde{\mathbf{e}}}{\sqrt{\text{Var}\left[\gamma\mathbf{u}^{\mathsf{T}}\boldsymbol{\Sigma}^{\frac{1}{2}}\mathbf{Q}\mathbf{V}_{\text{noise}}\tilde{\mathbf{e}}\right]}} \xrightarrow{\text{d}} \mathcal{N}(0, 1).$$

Remark also that

$$\text{Var}\left[\gamma\mathbf{u}^{\mathsf{T}}\boldsymbol{\Sigma}^{\frac{1}{2}}\mathbf{Q}\mathbf{V}_{\text{noise}}\tilde{\mathbf{e}}\right] = \frac{\gamma^2}{n}\text{tr}\left(\boldsymbol{\Sigma}\mathbf{Q}\mathbf{V}_{\text{noise}}\mathbf{V}_{\text{noise}}^{\mathsf{T}}\mathbf{Q}\right) \simeq \frac{\gamma^2}{n}\text{tr}\left(\boldsymbol{\Sigma}\mathbf{Q}\right)^2.$$

We thus obtain (17) in Corollary 1.

From (14) in Theorem 1, we have

$$\Pr(y'\hat{\boldsymbol{\beta}}^{\mathsf{T}}\mathbf{x}' > 0|(\mathbf{x}', y')) - \Pr(y'\tilde{\boldsymbol{\beta}}^{\mathsf{T}}\mathbf{x}' > 0|(\mathbf{x}', y')) \to 0.$$

Taking expectation over $(\mathbf{x}', y') \sim \mathcal{D}_{(\mathbf{x},y)}$, we get directly from the above equation that

$$\Pr(y'\hat{\boldsymbol{\beta}}^{\mathsf{T}}\mathbf{x}' > 0) - \Pr(y'\tilde{\boldsymbol{\beta}}^{\mathsf{T}}\mathbf{x}' > 0) \to 0.$$

It follows straightforwardly from (17) that

$$\Pr(y'\tilde{\boldsymbol{\beta}}^{\mathsf{T}}\mathbf{x}' > 0) - \Pr(yr > 0) \to 0,$$

leading to (18).

Similarly, we obtain (19) from (17) and (15), which concludes the proof.

### A.2.2 PROOF OF COROLLARY 2

It is easy to see that, when $e_1, \ldots, e_q$ are normally distributed, the random variable $r$ defined in (9) follows a Gaussian distribution $\mathcal{N}(m, \sigma^2 + \sum_{k=1}^{q}\psi_k^2)$. For $r \sim \mathcal{N}(m, \sigma^2 + \sum_{k=1}^{q}\psi_k^2)$ with $m, \sigma, \psi_1, \ldots, \psi_q$ as given in (13), we observe that the system of equations in (10) is invariant to the distributions of $e_1, \ldots, e_p$, thus yielding the same values of $\theta, \eta, \gamma, \omega_1, \ldots, \omega_q$, as well as the same $\kappa, m, \sigma^2, \psi_1, \ldots, \psi_q$.

Therefore, with Gaussian $e_1, \ldots, e_q$, $r$ follows a universal distribution $\mathcal{N}(m, \sigma^2 + \sum_{k=1}^{q}\psi_k^2)$ independent of the distributions of $e_{q+1}, \ldots, e_p$. We have also the same universality result on the distribution of $\text{prox}_{\kappa,\ell(\cdot,y)}(r)$ as the value of $\kappa$ is also insensitive to the distributions of $e_{q+1}, \ldots, e_p$.

Combining the above universal arguments on the distributions of $r$ and $\mathrm{prox}_{\kappa,\ell(\cdot,y)}(r)$ with Corollary 1, we prove that the Gaussian universality of in-distribution performance in Definition 3 holds if $e_1, \ldots, e_q$ are Gaussian variables.

It remains to demonstrate the breakdown of Gaussian universality on in-distribution performance if $e_1, \ldots, e_q$ are non-Gaussian.

Note first $\|\hat{\boldsymbol{\beta}}_{\ell,\lambda}\| = \Theta(1)$ with $\lambda = \Theta(1)$. The boundedness of $\|\hat{\boldsymbol{\beta}}_{\ell,\lambda}\|$ in the large $n, p$ limit is easily justified from the regularized optimization penalty (4) with $\lambda > 0$. Recall also

$$\lambda\hat{\boldsymbol{\beta}} = -\frac{1}{n}\sum_{i=1}^n \ell'(\hat{\boldsymbol{\beta}}^\mathsf{T}\mathbf{x}_i, y_i)\mathbf{x}_i,$$

from which we observe that, to ensure $\|\hat{\boldsymbol{\beta}}_{\ell,\lambda}\| = o(1)$, we need $\ell'(\hat{\boldsymbol{\beta}}^\mathsf{T}\mathbf{x}_i, y_i) = o(1)$. However when $\|\hat{\boldsymbol{\beta}}_{\ell,\lambda}\| = o(1)$, we have $\ell'(\hat{\boldsymbol{\beta}}^\mathsf{T}\mathbf{x}_i, y_i) \simeq \ell'(0, y_i) = \Theta(1)$. We thus get $\|\hat{\boldsymbol{\beta}}_{\ell,\lambda}\| = \Theta(1)$ by contradiction. Consequently, we have also $\|\hat{\boldsymbol{\beta}}_{\ell,\lambda}\| = \Theta(1)$ for the high-dimensional equivalent $\tilde{\boldsymbol{\beta}}$ given in Theorem 1. Since $\tilde{\boldsymbol{\beta}}^\mathsf{T}\mathbf{x}$ for $\mathbf{x} \sim \mathcal{D}_{(\mathbf{x},y)}$ independent of $\tilde{\boldsymbol{\beta}}$ has asymptotically the same distribution as $r$ in (9) according to Corollary 1, it follows from $\|\tilde{\boldsymbol{\beta}}_{\ell,\lambda}\| = \Theta(1)$ that $r = \Theta(1)$. Therefore $\eta = \mathbb{E}[h_\kappa(r, y)] = \Theta(1)$.

Let us reorganize the expression (16) as

$$\lambda\tilde{\boldsymbol{\beta}} = \eta\boldsymbol{\mu} + \sum_{k=1}^q \phi_k\mathbf{v}_k - \theta\mathbf{V}_{\mathrm{noise}}\mathbf{V}_{\mathrm{noise}}^\mathsf{T}\tilde{\boldsymbol{\beta}} + \gamma\boldsymbol{\Sigma}^{\frac{1}{2}}\mathbf{u}, \tag{61}$$

with

$$\phi_k = \omega_k - \theta\mathbf{v}_k^\mathsf{T}\mathbf{Q}\boldsymbol{\xi} = \mathbb{E}[h_\kappa(r, y)e_k], \quad \forall k \in \{1, \ldots, q\}.$$

Recall from (8) that

$$h_\kappa(t, y) = (\mathrm{prox}_{\kappa,\ell(\cdot,y)}(t) - t)/\kappa,$$

where $\mathrm{prox}_{\tau,f}(t) = \arg\min_{a\in\mathbb{R}}\left[f(a) + \frac{1}{2\tau}(a - t)^2\right]$ for $\tau > 0$ and convex $f \colon \mathbb{R} \to \mathbb{R}$. Due to the convexity of $\ell(\cdot, y)$ in Assumption 1, $h_\kappa(\cdot, y)$ is a decreasing function. As $e_1, \ldots, e_q$ are standardized variables of symmetric distribution according to Definition 1, we have

$$\mathbb{E}\left[h_\kappa(r, y)e_k \Big| \{e_d\}_{d\in\{1,\ldots,q\}\setminus k}, \tilde{e}, y\right] = \int_{-\infty}^{+\infty} h_\kappa\left(ym + \sigma\tilde{e} + \sum_{d\in\{1,\ldots,q\}\setminus k}\psi_d e_d + \psi_k e_k, y\right)e_k P_{e_k}(de_k),$$

where $P_{e_k}$ is the probability measure of $e_k$. As $e_k$ is a centered variable of symmetric probability distribution, we have

$$\int_{-\infty}^{+\infty} h_\kappa\left(ym + \sigma\tilde{e} + \sum_{d\in\{1,\ldots,q\}\setminus k}\psi_d e_d + \psi_k e_k, y\right)e_k P_{e_k}(de_k)$$

$$= \int_0^{+\infty} h_\kappa\left(ym + \sigma\tilde{e} + \sum_{d\in\{1,\ldots,q\}\setminus k}\psi_d e_d + \psi_k e_k, y\right)e_k P_{e_k}(de_k)$$

$$- \int_0^{+\infty} h_\kappa\left(ym + \sigma\tilde{e} + \sum_{d\in\{1,\ldots,q\}\setminus k}\psi_d e_d - \psi_k e_k, y\right)e_k P_{e_k}(de_k).$$

As $h_\kappa(t, y)$ decreases with $t$, there exists a positive value $a > 0$ such that

$$h_\kappa\left(ym + \sigma\tilde{e} + \sum_{d\in\{1,\ldots,q\}\setminus k}\psi_d e_d + \psi_k e_k, y\right) - h_\kappa\left(ym + \sigma\tilde{e} + \sum_{d\in\{1,\ldots,q\}\setminus k}\psi_d e_d - \psi_k e_k, y\right) = -2a\psi_k e_k.$$

In the end, we have

$$\phi_k = \mathbb{E}[h_\kappa(r, y)e_k] = -\alpha_k\psi_k$$

where $\alpha_k > 0$ for all $k \in \{1, \ldots, q\}$.

Plugging in the above expression of $\phi_k$, we rewrite (61) as

$$\lambda\tilde{\boldsymbol{\beta}} = \eta\boldsymbol{\mu} - \sum_{k=1}^{q} \alpha_k\psi_k\mathbf{v}_k - \theta\mathbf{V}_{\mathrm{noise}}\mathbf{V}_{\mathrm{noise}}^{\mathsf{T}}\tilde{\boldsymbol{\beta}} + \gamma\boldsymbol{\Sigma}^{\frac{1}{2}}\mathbf{u}.$$

Taking expectation at the both sides of the above equation, we get

$$\lambda\mathbf{Q}\boldsymbol{\xi} = \eta\boldsymbol{\mu} - \sum_{k=1}^{q} \alpha_k\psi_k\mathbf{v}_k - \theta\mathbf{V}_{\mathrm{noise}}\mathbf{V}_{\mathrm{noise}}^{\mathsf{T}}\tilde{\boldsymbol{\beta}}.$$

Therefore

$$\lambda\mathbf{V}_{\mathrm{info}}^{\mathsf{T}}\mathbf{Q}\boldsymbol{\xi} = \lambda\boldsymbol{\psi} = \eta\mathbf{V}_{\mathrm{info}}^{\mathsf{T}}\mathbf{V}_{\mathrm{info}}\mathbf{s} - \sum_{k=1}^{q} \mathbf{V}_{\mathrm{info}}^{\mathsf{T}}\mathbf{V}_{\mathrm{info}}\mathrm{diag}(\alpha_1,\ldots,\alpha_q)\boldsymbol{\psi},$$

where $\mathbf{V}_{\mathrm{info}} = [\mathbf{v}_1,\ldots,\mathbf{v}_q]$, $\boldsymbol{\psi} = [\psi_1,\ldots,\psi_q]^{\mathsf{T}}$ and $\mathbf{s} = [s_1,\ldots,s_q]$. As $\mathbf{V}_{\mathrm{info}}$ and $\mathbf{s}$ are both deterministic with no presumed relation, we consider them to independent in some probability space. We obtain thus

$$\boldsymbol{\psi} = \eta\left(\lambda\mathbf{I}_q + \mathbf{V}_{\mathrm{info}}^{\mathsf{T}}\mathbf{V}_{\mathrm{info}}\mathrm{diag}(\alpha_1,\ldots,\alpha_q)\right)^{-1}\mathbf{V}_{\mathrm{info}}^{\mathsf{T}}\mathbf{V}_{\mathrm{info}}\mathbf{s} = \Theta(1).$$

Therefore $r$ is non-Gaussian unless in the case of Gaussian $e_1,\ldots,e_q$, leading to the breakdown of in-distribution performance in Definition 3.

### A.2.3 PROOF OF COROLLARY 3

As discussed in Appendix A.2.2, the system of equations in (10), which determines the distribution of $\tilde{\boldsymbol{\beta}}$, is universal in the case of normally distributed $e_1,\ldots,e_q$. We obtain directly the Gaussian universality of classifier in Definition 3 under the condition of Gaussian $e_1,\ldots,e_q$

Note importantly that when $\partial\ell(\hat{y},y)/\partial\hat{y}$ is a linear function of $\hat{y}$ of form

$$\partial\ell(\hat{y},y)/\partial\hat{y} = a\hat{y} + b(y)$$

for some constant $a > 0$ (due to the convexity of $\ell(\cdot,y)$) and $b(y)$ independent of $\hat{y}$, we have

$$\mathrm{prox}_{\kappa,\ell(\cdot,y)}(\hat{y}) = \frac{\hat{y} - \kappa b(y)}{1 + \kappa a},$$

which leads to

$$h(\hat{y},y) = \frac{\mathrm{prox}_{\kappa,\ell(\cdot,y)}(\hat{y}) - \hat{y}}{\kappa} = \frac{-a\hat{y} - b(y)}{1 + \kappa a}.$$

Recall from (9) that

$$r = ym + \sigma\tilde{e} + \sum_{k=1}^{q} \psi_k e_k.$$

The equations in (10) thus become

$$\theta = -\mathbb{E}[\partial h_\kappa(r,y)/\partial r] = \frac{a}{1 + \kappa a}, \quad \eta = \mathbb{E}[yh_\kappa(r,y)] = \frac{-am - \mathbb{E}[yb(y)]}{1 + \kappa a},$$

$$\gamma = \sqrt{\mathbb{E}[h_\kappa(r,y)^2]} = \frac{\sqrt{a^2\left(m^2 + \sigma^2 + \sum_{k=1}^{q}\psi^2\right) + \mathbb{E}[b(y)^2] - 2am\mathbb{E}[yb(y)]}}{1 + \kappa a}$$

$$\omega_k = \mathbb{E}[h_\kappa(r,y)e_k] + \theta\cdot\mathbf{v}_k^{\mathsf{T}}\mathbf{Q}\boldsymbol{\xi} = \frac{-a\psi_k}{1 + \kappa a} + \theta\cdot\mathbf{v}_k^{\mathsf{T}}\mathbf{Q}\boldsymbol{\xi},$$

which are independent of the distributions of the noise variables $e_1,\ldots,e_p$. We prove thus the Gaussian universality of classifier in Definition 3 when $\partial\ell(\hat{y},y)/\partial\hat{y}$ is a linear function of $\hat{y}$.

Conversely, when $\partial\ell(\hat{y},y)/\partial\hat{y}$ is a nonlinear function of $\hat{y}$, $h_\kappa(r,y)$ is also a nonlinear function of $r$. Consequently, the values of $\theta,\eta,\gamma,\omega_1,\ldots,\omega_q$ depend on the higher-order moments of $e_1,\ldots,e_q$ besides the first two, thus leading to the breakdown of Gaussian universality on classifier in the presence of non-Gaussian $e_1,\ldots,e_q$.

## B  EXPERIMENTS ON REAL DATA

In this section, we report experimental results on Fashion-MNIST image data (Xiao et al., 2017) to show how the conditions of Gaussian universality provided in Corollaries 2 and 3 can be used to *understand and predict* Gaussian universality phenomena on real data learning problems.

We have discussed two types of universality in this paper: universality on **in-distribution performance** and universality on **classifier** (see Definition 3 for more details). To discuss these two types of universality, we distinguish, depending on the Gaussianity of informative factors and the use of square loss, the following three scenarios:

1. **Scenario 1**: in the case of non-Gaussian informative factors and when a non-square loss is used, *neither* the universality on in-distribution *nor* the universality on classifier holds;

2. **Scenario 2**: in the case of non-Gaussian informative factors and when a square loss is used, the universality on in-distribution breaks down while the universality on classifier still holds;

3. **Scenario 3**: in the case of Gaussian informative factors and when an arbitrary (square or non-square) loss is used, *both* the universality on in-distribution *and* the universality on classifier hold.

To see if these three scenarios derived under LFMM can be "reproduced" on realistic Fashion-MNIST image data, we conduct first a principle component analysis (PCA) on standardized Fashion-MNIST data to extract the informative factors. To obtain the equivalent GMM in Definition 2 for a mixture of Fashion-MNIST data, we estimate the class mean and the class covariance for each class of Fashion-MNIST, using all available samples in that class.

Here, we consider the following two cases to illustrate the different effects of Gaussian and non-Gaussian informative factors on ERM classification:

1. **Case 1**: Classes 4&5 of Fashion-MNIST data, as an example of *non-Gaussian* informative factors; and

2. **Case 2**: Classes 3&7 of Fashion-MNIST data, for which *approximately Gaussian* informative factors can be observed.

In Figure 4 and Figure 5 we compare, for the aforementioned two cases, the (empirical) distributions of the first two informative factors obtained from PCA. We observe that the informative factors of Classes 4&5 have highly asymmetric distributions, corresponding to a strong deviation from the Gaussianity, while the distribution of informative factors in Classes 3&7 are much closer to the form of normal density function in comparison.

Figures 6 to 10 then provide empirical results on these two cases to demonstrate the universal or non-universal behavior with respect to the in-distribution performance and the ERM classifier, under the three scenarios on informative factors and loss function listed at the beginning of this section.

We discuss first Scenario 1 with non-Gaussian informative factors (Case 1 on Classes 4&5) and non-square losses. Under this scenario, the in-distribution performance is predicted, as per Corollary 2, to be different from that under the equivalent GMM, as can be observed in the middle and right plots of Figure 7. According to Corollary 3, the universality on classifier does *not* hold in this case either. In other words, the classifier trained on Fashion-MNIST data and the one trained on data drawn from the equivalent GMM give different performances on the *same* test Fashion-MNIST data. This is empirically manifested in middle and right plots of Figure 9 and suggests an effective learning using non-square losses from high-order moment information beyond the class mean and covariance.

It is interesting to compare Scenario 1 with Scenario 2, where we use square and non-square losses on non-Gaussian informative factors. In Scenario 2, while we still do *not* have a universal in-distribution performance as evidenced in the left column of Figure 7, the classifier trained on equivalent GMM data gives practically the *same* performance on test Fashion-MNIST data as the classifier trained on realistic Fashion-MNIST data, as shown in the left plot of Figure 9. This means that the square loss *fails* to learn from Fashion-MNIST data beyond the information contained in the equivalent GMM (i.e., the class mean and covariance). Consequently, the square loss yields suboptimal performance as shown in the left plot of Figure 6.

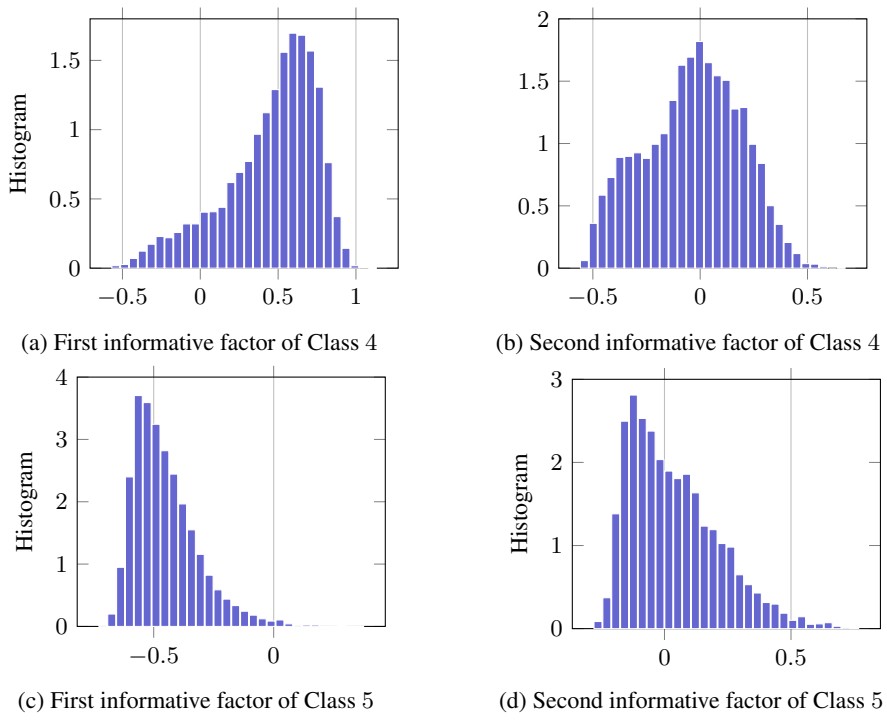

(a) First informative factor of Class 4

(b) Second informative factor of Class 4

(c) First informative factor of Class 5

(d) Second informative factor of Class 5

Figure 4: Histogram of the first and second information factors of Class 4 and 5, estimated using all samples from the Fashion-MNIST dataset.

In Scenario 3 with (approximately) Gaussian informative factors (Case 2 on Classes 3&7), the two types of Gaussian universality (on in-distribution performance and on classifier) hold for any choice of loss function. The universality on in-distribution performance is demonstrated in Figure 8, where we observe a much closer match between the in-distribution performance on Fashion-MNIST and the equivalent GMM, in comparison with the drastically different in-distribution performances reported in Figure 7 for Case 1 on Classes 4&5 with non-Gaussian informative factors. The universality on classifier that holds for any loss (square or non-square) in this scenario is empirically confirmed by comparing Figure 10 with Figure 9. Otherwise speaking, in this case, Fashion-MNIST data are treated by the ERM classifier *as if they were Gaussian mixture data*. As a result, we predict that there is little room in trying to do better than the square loss (that is identified by Taheri et al. (2021b); Mai & Liao (2019) to be the optimal loss under GMM). This is consistent with the empirical performances given by different losses displayed in the right plot of Figure 6.

As a side note, the empirical universal results provided in Figure 2 of (Dandi et al., 2024) also involved Fashion-MNIST data. However as Dandi et al. (2024) trained the classifier on *synthetic data* generated by a conditional GAN learned from the Fashion-MNIST dataset, these experimental results are not directly comparable to ours, which were obtained from a direct training on Fashion-MNIST data.

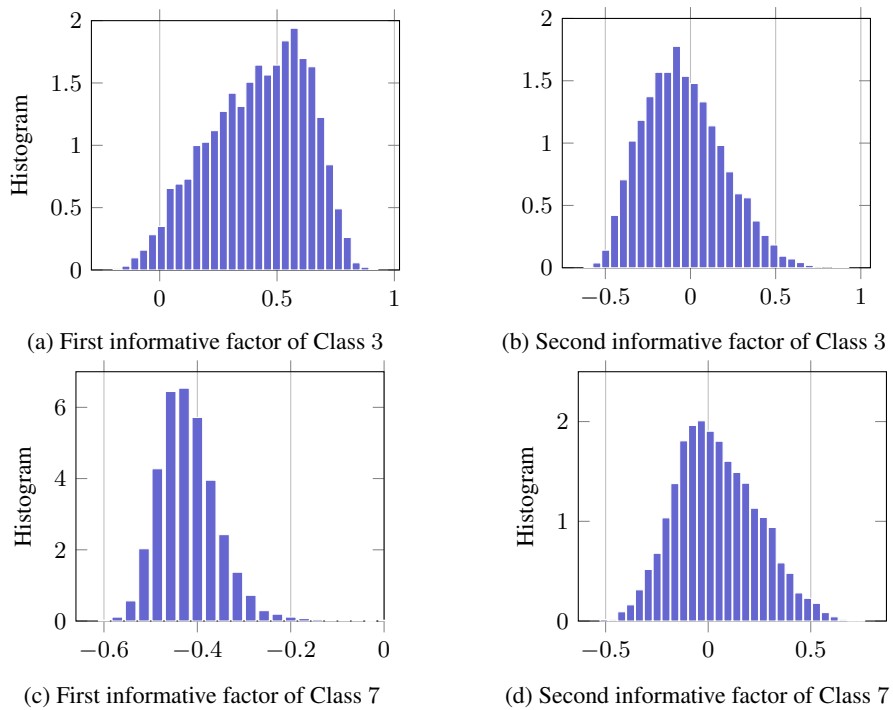

Figure 5: Histogram of the first and second information factors of Class 3 and 7, estimated using all samples from the Fashion-MNIST dataset.

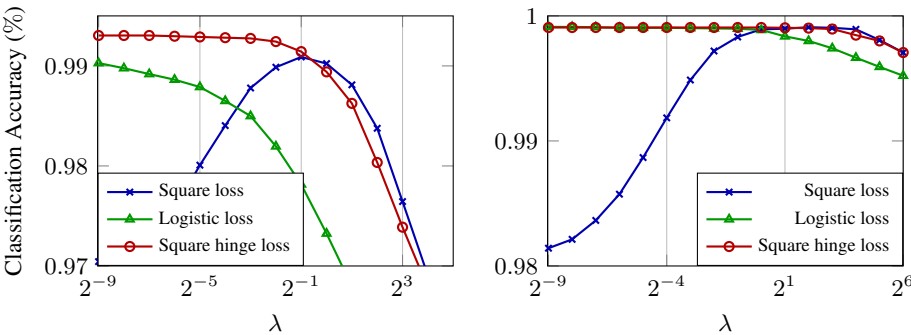

Figure 6: Classification accuracies as a function of the regularization penalty, for square, logistic, and square hinge loss, on Fashion-MNIST data of sample size $n = 512$. **Left**: Class 4 versus 5, as an example of non-Gaussian information factors showed in Figure 4. **Right**: Class 3 versus 7, as an example of (close-to) Gaussian information factors showed in Figure 5.

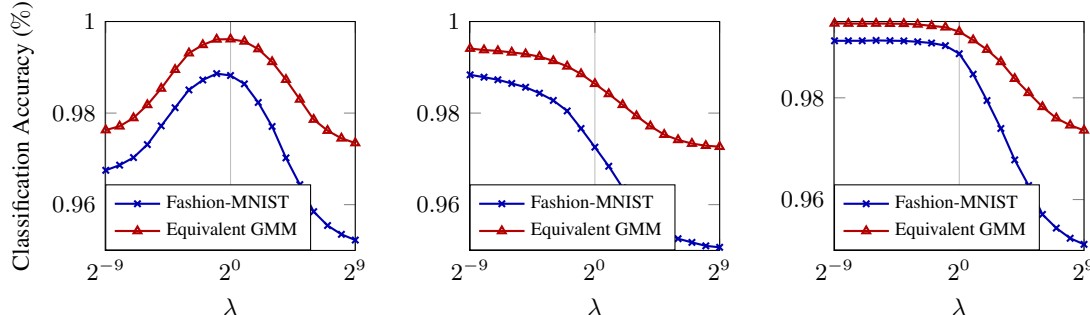

Figure 7: **In-distribution** classification accuracies as a function of the regularization penalty $\gamma$, for Fashion-MNIST data (Class 4 versus 5, as an illustrating example of non-Gaussian informative factors as shown in Figure 4) and Equivalent GMM of sample size $n = 512$, with square (**left**), logistic (**middle**), and square hinge (**right**) losses.

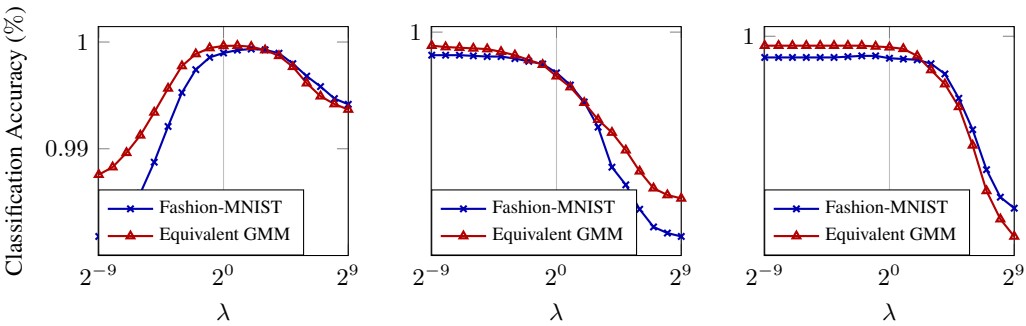

Figure 8: **In-distribution** classification accuracies as a function of the regularization penalty $\gamma$, for Fashion-MNIST data (Class 3 versus 7, as an illustrating example of close-to-Gaussian informative factors as shown in Figure 5) and Equivalent GMM of sample size $n = 512$, with square (**left**), logistic (**middle**), and square hinge (**right**) losses.

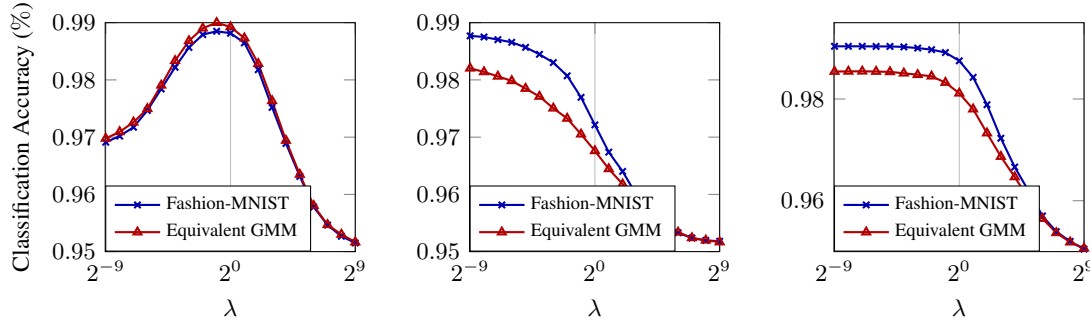

Figure 9: Classification accuracies on test Fashion-MNIST data as a function of the regularization penalty $\gamma$, given by classifiers trained on Fashion-MNIST data (Class 4 versus 5, as an illustrating example of non-Gaussian informative factors as shown in Figure 4) and on Equivalent GMM data of sample size $n = 512$, with square (**left**), logistic (**middle**), and square hinge (**right**) losses.

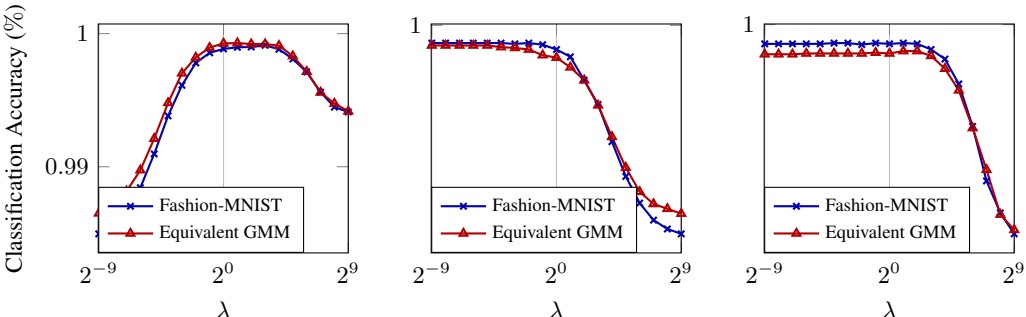

Figure 10: Classification accuracies on test Fashion-MNIST data as a function of the regularization penalty $\gamma$, given by ERM classifiers trained on Fashion-MNIST data (Class 3 versus 7, as an illustrating example of close-to-Gaussian informative factors as shown in Figure 5) on Equivalent GMM data of sample size $n = 512$, with square (**left**), logistic (**middle**), and square hinge (**right**) losses.

