# OpenReview forum: "The Breakdown of Gaussian Universality in Classification of High-dimensional Linear Factor Mixtures"
_ICLR.cc/2025/Conference — ICLR 2025 Poster_

### Official Review · Reviewer_66dx · 2024-10-20

**Soundness:** 3
**Presentation:** 3
**Contribution:** 3
**Rating:** 6
**Confidence:** 3

**Summary:**

This paper studies the Gaussian universality and its breakdown in classification problems in the setting of high-dimensional linear factor mixture models (LFMM). LFMMs is an extension of the Gaussian Mixture Model in the classification problem. The study demonstrates that the Gaussian universality does not hold for LFMM in general. Furthermore, the paper considers the relationship with the squared loss and Gaussian universality, and clarify the importance of selecting an appropriate loss function. The theoretical analysis provides a new perspective on high-dimensional data analysis. Several numerical experiments are conducted to confirm the theoretical findings based on asymptotic theory. More comprehensive numerical experiments and verification using real data are needed.

**Strengths:**

- Regarding the breakdown of Gaussian universality:
Many previous studies have investigated data that follow a Gaussian distribution and scenarios in which the Gaussian universality holds. Since these assumptions do not necessarily hold for real data, it is necessary to analyse more general situations. In the paper, the authors reveal the conditions under which the Gaussian universality is violated for LFMMs. The results can provide new insights into high-dimensional data analysis.

- Evaluation of the optimality of the squared loss
It is known that the squared loss is the optimal loss function under the Gaussian mixture models. In contrast to that, this paper discusses the possibility that the squared loss is sub-optimal under a more general LFMMs. It suggests that existing works do not necessarily apply in cases where the Gaussian universality breaks down, and presents a new challenge of selecting an appropriate loss function.

- Numerical experiments:
In order to verify the theoretical results, several numerical experiments are demonstrated. These experiments examine the behaviour of the classifier when the Gaussian universality is violated. In Figure 2, the performance of the square loss and the square hinge loss is compared, and one can see that the numerical result is consistent with the theoretical results. Similarly, Figure 3 suggests that the squared loss may not be the optimal solution, and this is also consistent with the theoretical results.

**Weaknesses:**

This paper deals with the important problem of the breakdown of Gaussian universality in the classification problem of high-dimensional LFMMs. The following are some of the concerns that should be considered.

- In some theorems and corollaries, is it possible to evaluate the convergence speed? Although the proof in the supplementary material seems to evaluate the convergence speed, it would be better for the reader to understand the paper more deeply if the speed were explicitly stated in the main text.

- Theoretical claims are supported by some numerical experiments. However, these experiments are only verified using synthetic data. It is unclear how much the real-world data deviates from the Gaussian universality. Additional experiments using real data are needed to strengthen the paper's arguments. In addition, in line 53 of the introduction, the possibility of observing Gaussian universality in real data is mentioned. I think a supplementary explanation is needed to relate this statement to the results of the paper.

- This paper clarifies the conditions under which the Gaussian universality breaks down, but does not discuss the extent of this breakdown in detail. Can the extent of the deviation from the Gaussian universality be evaluated as a statistical bias? If the impact of this bias on learning performance can be quantified, it would be very useful in practical applications.

**Questions:**

See Strengths and Weaknesses.

---

> ### Author Response · Authors · 2024-11-20
>
> We thank the reviewer for his/her detailed read of the paper and insightful comments that help us improve this paper.
>
>
> **Convergence speed**:
>
> As pointed out by the reviewer, it can be derived from our proof in Appendix that the differences in expectation provided in Theorem 1 (and thus the generalization and training performances per Corollary 1)  converge at a speed of at least $O(p^{-1/4})$.
> Nonetheless, we believe that this rate is largely sub-optimal.
> As many high-dimensional asymptotic analyses of ML methods (cited in the introduction of the manuscript), the primary focus of this paper is to derive the system of equations that determines the asymptotic performance under LFMM as $n,p \to \infty$ together, we did **not** make the effort to obtain the optimal convergence speed. That is why we chose not to explicitly state our convergence speed in the main text, as it is likely to be overly pessimistic. Instead, we provided empirical validation of our asymptotic results by showing a close match in practice for $n,p$ only in a few hundreds.
>
>
>
> **Real data experiments**:
>
> We agree that it would be interesting to test to what extent our conclusions on the Gaussian universality breakdown apply in real data learning scenarios.
> We are running additional experiments on real-world datasets, particularly to highlight:
> - the impact of non-Gaussian informative factors on the Gaussian universality breakdown, as suggested in our Corollary 2; and
> - the robustness of square loss to the Gaussian universality breakdown, as discussed in our Remark 2.
> We will post these numerical results as soon as  they are ready.
>
> As for the inquiry of the reviewer about the empirical results mentioned in Line 53 of the submission, Sur & Candès, (2019); Taheri et al. (2021b) only tested the Gaussian universality on synthetic data in a regression setting, which is not directly related to the classification problem of mixture data under study in our paper. Loureiro et al. (2021) used a random feature mapping on MNIST and fashion-MNIST data to induce the conditional one-directional CLT in (3), so as to observe Gaussian universality on these image data. We believe that the random feature mapping kind of destroyed the structure of non-Gaussian informative factors which might be present in the original data vectors. It is indeed interesting to discuss these empirical results with respect to ours (we will add this discussion along with the experiments on real data).
>
>
> **Deviation from Gaussian universality**:
>
> A benefit of our exact performance characterization under LFMM provided in Theorem 1 and Corollary 1 is that it can be used to assess quantitatively the deviation from Gaussian universality. In particular, it can be shown from our Theorem 1 that under GMM, the "high-dimensional equivalent $\tilde{\boldsymbol{\beta}}$" of the ERM solution $\hat{\boldsymbol{\beta}}$ is aligned in expectation with $(\lambda\mathbf{I}+\theta\boldsymbol{\Sigma})^{-1}\boldsymbol{\mu}$. In contrast, under LFMM, $\mathbb{E}[\tilde{\boldsymbol{\beta}}]$ lies in the direction of $(\lambda\mathbf{I}+\theta\boldsymbol{\Sigma})^{-1} (\boldsymbol{\mu}+  \sum_{k=1}^q \omega_{k} \mathbf{v}_k)$, where $\omega_1,\ldots,\omega_q\neq 0$ for non-Gaussian informative factors $z_1,\ldots, z_q$, causing $\mathbb{E}[\tilde{\boldsymbol{\beta}}]$ to be more or less aligned with the directions $\mathbf{v}_1,\ldots, \mathbf{v}_q$ controlled by the non-Gaussian informative factors.
>
> However the value of $\omega_k$ can not be assessed without knowing the distributions of the non-Gaussian informative factor $z_k$. Therefore before applying our results to assess the deviation from Gaussian universality, it requires a pre-processing step to estimate the LFMM from the data set.
>
> We would not call the deviation from Gaussian universality as statistical bias, as it reflects the adaptation of the ERM classifier to non-Gaussian informative factors, which is most likely to be **beneficial** to the classification performance.

---

> > ### Comment · Reviewer_66dx · 2024-11-23
> >
> > Thank you very much for the detailed response. I will retain my score.

---

> > > ### Author Response · Authors · 2024-11-25
> > >
> > > We thank the reviewer for reading our reply. As promised, we provided additional experiments on real (Fashion-MNIST) data to test the applicability of our universality conditions on informative factors and loss function. As announced in the second general reply, these results were reported in Appendix B of the (newest) revised version of the manuscript.
> > >
> > > We also incorporated the comments on classifier bias (discussed in the last point of our first rely) in Remark 1 of the revised manuscript.
> > >
> > > Thanks again to the reviewer for the helpful suggestions.

---

### Official Review · Reviewer_8nd6 · 2024-10-25

**Soundness:** 2
**Presentation:** 3
**Contribution:** 2
**Rating:** 6
**Confidence:** 4

**Summary:**

Given the relevance of Gaussian or Gaussian mixtures for dataset modeling in theoretical machine learning, a recent line of works investigated under which conditions the Gaussian assumption can be justified in the high-dimensional regime. This line of works resulted in a series of contributions showing that a *Gaussian Universality Principle* can be indeed safely invoked in many cases. To show the limitations of such an assumption, the authors propose a classification task on a linear factor mixture model, in which two clouds in a $p$-dimensional space have to be classified. The clouds are constructed by using a linear factor model each so that the signal (i.e., the cloud mean) is determined by $q<p$ independent directions only. It is then proven that in this setting Gaussian universality might break down, and the statistical properties of the estimators obtained by convex ERM are asymptotically characterized.

**Strengths:**

The paper presents an interesting model, that is completely characterizable in the asymptotic regime, to test and break down Gaussian universality (GU). Amongst the relevant features observed by the authors, the directions in which the signal is absent play no role in the statistics of the preactivations (i.e., the in-distribution performance), and square loss exhibits special robustness with respect to GU. The results in the paper are rigorously proven and the problem is attacked by adopting a leave-one-out strategy. Proofs are given in the Appendix. Finally, I found both the manuscript and the appendix very clear and well-written.

**Weaknesses:**

The paper claims to be ```the first to characterize the learning performance when the Gaussian universality``` (GU) ```breaks down```. However, in one of the cited papers by [Adomaityte et al. (2024)](https://proceedings.neurips.cc/paper_files/paper/2023/hash/88be023075a5a3ff3dc3b5d26623fa22-Abstract-Conference.html) a similar (in spirit) analysis has been recently performed: the classification problem of two non-Gaussian clouds has been analyzed by using the heuristic replica method, analytically showing that Gaussian universality can indeed break down quite simply: the statistical properties of the estimators are also given therein. Although the type of dataset model adopted is in general different, my impression is that the results are on the same line as the one published in this manuscript: unless I am mistaken, the LFMM includes as special case a subfamily of the setup considered therein. If for example $\mathbf V=\mathbf I_p$ and $e_k=a\eta_k$ with $\eta_k\sim\mathcal N(0,1)$ and $a$ positive random quantity having $\mathbb E[a^2]<+\infty$, one recovers the finite-variance model in [Adomaityte et al. (2024)](https://proceedings.neurips.cc/paper_files/paper/2023/hash/88be023075a5a3ff3dc3b5d26623fa22-Abstract-Conference.html) where GU breaking is observed (note however that in this model $\mathbb E[a^4]=\mathbb E[e_i^2e_j^2]\neq \mathbb E[e_i^2]\mathbb E[e_j^2]=\mathbb E[a^2]^2$ in general).

One might argue that, beyond classification, similar Gaussian universality-breaking results were present in the line of works of [El Karoui et al. (2013)](https://www.pnas.org/doi/10.1073/pnas.1307842110) on regression (see also a different paper by [Adomaityte et al. (2024)](https://arxiv.org/pdf/2309.16476)) although the focus in these works was more on robustness. One weakness of the paper is therefore that it might appear as a complement to the existing contributions in which a lack of GU is observed and characterized in the same asymptotic regime.

**Questions:**

I list below a series of observations and questions.

- I found a little bit misleading the sentence at lines 139-141: the one dimensional CLT in Eq. (2) is independent of any ERM solution $\hat{\boldsymbol\beta}$.
- The notation in Eq. (3) is not very transparent with respect to the average over $\mathbf g$. It would be maybe more clear to just consider $\mathbb E[f(\mathbf g^\top\boldsymbol\beta)]$ and then specify that $\mathbf g\sim\mathcal N(\mathbb E[\mathbf x|y_{\mathbf x}=C],\mathrm{Cov}[\mathbf x|y_{\mathbf x}=C])$.
- In the discussion of the elliptical distribution case, the quantity $a$ is random (this should maybe specified when introduced). It is not clear to me what the authors refer to when commenting about the application of the CGEP to this case: from what I understand *each data point in the dataset* has a different $a$ from a given distribution, so for example in the paper by [Adomaityte et al. (2024)](https://proceedings.neurips.cc/paper_files/paper/2023/hash/88be023075a5a3ff3dc3b5d26623fa22-Abstract-Conference.html) conditioning on the class does not lead to a distribution that satisfies a CGEP, so elliptic distributions can be seen as another example of a simple setup that ``breaks'' Gaussian universality (which is I think the point of the cited paper).
- From what I understand, the noise terms in Eq. (5) are supposed to be independent: is this correct?
- The possibility of accessing an explicit form for $\tilde{\boldsymbol\beta}$ in Eq. (16) strongly depends on the fact that ridge regularization has been chosen. Unless I am mistaken, it might be worth clarifying this point.
- Is the discrepancy appearing in Fig 1 middle due to finite-size effects? In this case, is this numerically verifiable? Also, in the label, I suppose that 'normal' has to follow 'Middle'.
- Corollary 3 and the following Remark 2 are interesting results, although I was wondering how much they depend on the specific LFMM form of the dataset. Again, in the aforementioned paper by [Adomaityte et al. (2024)](https://proceedings.neurips.cc/paper_files/paper/2023/hash/88be023075a5a3ff3dc3b5d26623fa22-Abstract-Conference.html) a breaking of Gaussian universality is indeed observed also in the classification of two elliptic clouds *with square loss* (see Figure 1 therein). My impression therefore is that the 'robustness' of square loss with respect to GU in Corollary 3 might stem more from the specific model considered in the paper than from some intrinsic property of square loss.
- In line 535, ```anlaysis``` to be changed in ```analysis```.
- In the Appendix, at line 870 what does the notation $\partial_-$ mean? Also, lines 887-888 should be checked. A square bracket is missing at the end of line 1075. At line 1131, $E$ has to be replaced with $\mathbb E$. I think an expectation has to be added under the square root in the definition of $\gamma$ at line 1395.

---

> ### Author Response · Authors · 2024-11-20
>
> We thank the reviewer for the careful reading of our paper and the helpful suggestions.
>
> **With respect to other non-universality results**
>
> We are aware of the existence of other forms of Gaussian universality breakdown, such as the one induced by elliptically distributed data characterized in the work of [Adomaityte et al. (2024)]( https://proceedings.neurips.cc/paper_files/paper/2023/hash/88be023075a5a3ff3dc3b5d26623fa22-Abstract-Conference.html). In our opinion, the non-universality result in [Adomaityte et al. (2024)]( https://proceedings.neurips.cc/paper_files/paper/2023/hash/88be023075a5a3ff3dc3b5d26623fa22-Abstract-Conference.html) is fundamentally different from ours for several reasons stated below.
> - Due to the existence of the scaling variable $a$ in elliptically distributed data $\mathbf{x}_i$, it is easy to see that the conditional one-directional CLT in (3) **cannot** hold (unless one considers the expectation conditioned on *both* the label $y$ and the scaling variable $a$), consequently the Gaussian universality is bound to break down. In contrast, it is not clear whether the conditional one-directional CLT in (3) holds for the linear factor mixture model (LFMM) under study *unless a precise performance analysis is performed*.
> - Contrarily to the constant breakdown of Gaussian universality under the elliptically distributed data setting in [Adomaityte et al. (2024)]( https://proceedings.neurips.cc/paper_files/paper/2023/hash/88be023075a5a3ff3dc3b5d26623fa22-Abstract-Conference.html), our analysis revealed that the Gaussian universality **can sometimes hold** under LFMM. By identifying conditions of Gaussian universality under LFMM, our analysis allows a deeper understanding on the applicability of Gaussian universality and the causes of its breakdown.
> - Another major difference between LFMM and the elliptical data setting considered in [Adomaityte et al. (2024)](https://proceedings.neurips.cc/paper_files/paper/2023/hash/88be023075a5a3ff3dc3b5d26623fa22-Abstract-Conference.html) lies in the fact that the concentration of quadratic forms of $\mathbf{x}_i$ in (1) does not hold for elliptical data, as opposed to LFMM.
> This is also the cause behind their different behaviors regarding the robustness of square loss with respect to Gaussian universality pointed out by the reviewer.
> As we know, the ERM solution given by the square loss is a weighted mean of the training data vectors $\mathbf{x}_i$ multiplied by the inverse of the sample covariance matrix (SCM) of $\mathbf{x}_i$.
> As the scaling variable $a$ breaks the concentration of quadratic forms of $\mathbf{x}_i$ in (1), the universality of the SCM collapses (which was briefly discussed in Lines 161-164 of the submission; in fact, the covariance considered in Adomaityte et al. (2024) can be unbounded), and so does the universality of ERM with square loss.
>
> To avoid ambiguity regarding the positioning of the present work with respect to other non-universality results, we have removed the expression in question "*the first to characterize the learning performance when the Gaussian universality breaks down*" in the concluding remarks of the manuscript and modified the section accordingly.
> We have also added a phrase in Line 172 of the revision to comment on the non-universal behavior of elliptically distributed data induced by the scaling variable $a$.
>
> **Further comparison to the work of Adomaityte et al. (2024)**
>
> - This is **no** overlap between our LFMM and the elliptically distributed data setting considered in [Adomaityte et al. (2024)](https://proceedings.neurips.cc/paper_files/paper/2023/hash/88be023075a5a3ff3dc3b5d26623fa22-Abstract-Conference.html) for classification and in [Adomaityte et al. (2024)](https://arxiv.org/pdf/2309.16476)  for regression.
> As explained earlier, the scaling variable $a$ breaks the concentration of the norm of elliptically distributed $x_i$. On the other hand, it is easy to check that under our LFMM, the norm of $x_i$ always concentrates as a consequence of the statistical independence of noises $e_1,\ldots,e_p$ by the law of large numbers.
> - From a technical perspective, our analysis also differs from [Adomaityte et al. (2024)](https://proceedings.neurips.cc/paper_files/paper/2023/hash/88be023075a5a3ff3dc3b5d26623fa22-Abstract-Conference.html). Precisely, [Adomaityte et al. (2024)](https://proceedings.neurips.cc/paper_files/paper/2023/hash/88be023075a5a3ff3dc3b5d26623fa22-Abstract-Conference.html) used the replica heuristic to find the asymptotic equations on the learning performance, whereas our results are proven through a series of concentration results derived with the help of leave-one-out manipulation.

---

> > ### Author Response · Authors · 2024-11-20
> >
> > **Reply to the list of questions**
> >
> > - For the Gaussian equivalent principle (GEP) to be valid, the one-dimensional CLT in (2) has to hold in a subspace containing the solution $\hat{\boldsymbol{\beta}}$. Let us consider an extreme case where all elements of $\hat{\boldsymbol{\beta}}$ are zero except the first one. Then if the fist element of $\mathbf{x}$ follows a uniform distribution, the one-dimensional CLT in (2) will certainly fail at the point $\boldsymbol{\beta}=\hat{\boldsymbol{\beta}}$ despite the statistical independence between $\mathbf{x}$ and $\hat{\boldsymbol{\beta}}$, as $\mathbf{x}^{\sf T}\hat{\boldsymbol{\beta}}$ is uniformly distributed. This is what we meant to convey in Lines 139-141 of the submission.
> > - We thank the reviewer for the suggestion, which has been adopted in the revised version of the paper.
> > - We now explicitly describe $a$ as a scaling **random** variable at Line 161 of the revision. As explained in Lines 166-168 of the submission, if not conditioned on $a$, neither GEP nor CGEP will hold for elliptical data. As we said in Lines 170-172 of the submission,  [Adomaityte et al. (2024)](https://proceedings.neurips.cc/paper_files/paper/2023/hash/88be023075a5a3ff3dc3b5d26623fa22-Abstract-Conference.html) “provided the asymptotic classification error depending on the distribution of $a$”, which implies a non-universal performance that varies with the distribution of $a$. We have modified the phrase to make the message clearer in the revised version.
> > - The noises $e_1,\ldots,e_p$ in (5) are indeed independent as the factors $z_1,\ldots,z_p$ are independent according to Line 202 of the submission. In the revised version, we explicitly state the independence of $e_1,\ldots,e_p$ to avoid confusion.
> > - We are not sure if we understood well this question. According to Theorem 1, $\tilde{\boldsymbol{\beta}}$ is a "high-dimensional equivalent" of $\hat{\boldsymbol{\beta}}$, which is defined as the solution to the ridge-regularized ERM in (4).
> > - As suspected by the reviewer, the discrepancy is indeed due to the fine-size effect in simulations. The match with the theory is, in our view, satisfactory considering that the displayed histograms are obtained from a single training of the ERM solution at $n,p$ only in a few hundreds. Thanks for pointing out the typo, "normal" should indeed follow "Middle".
> > - Please refer to the third point of **With respect to other non-universality results** above.
> > - Thank you for spotting the typo.
> > - The notation $\partial_{-}$ at Line 870 stands for the left derivative, namely $\ell''(t,y)=\frac{\partial_{-}\ell’(t,y)}{\partial_{-} t}=\lim_{a\to t^{-}}\frac{\ell’(a,y)-\ell’(t,y)}{a-t}$. There exist indeed some editing errors in Lines 887-888, which have been corrected in the revision, along with other typos found by the reviewer. Many thanks to the reviewer for helping us identify the typos. We appreciate the thorough reading you gave to our paper.

---

> > > ### Comment · Reviewer_8nd6 · 2024-11-23
> > >
> > > I would like to thank the authors for their reply and for clarifying my doubts.
> > >
> > > I acknowledge the fact that the model presented in this paper is (to my knowledge) new, and has the benefits of some special properties. The reason for my (mildly) negative score is that I am not sure if this model leads to relevant new insights on GUs. One might argue that the fact that GUs cannot hold in general in the considered asymptotic regime has been already established. Sufficient conditions for its validity have been identified in many works cited by the authors, and the *necessity* of some of such conditions, e.g. the so-called pointwise normality, was pointed out as well in the work of [Montanari and Saeed](https://arxiv.org/abs/2202.08832), where a simple counterexample with $\boldsymbol x_i\in\mathrm{Unif}(\\{-1,1\\}^p)$ was used (so that by construction $\\|\boldsymbol x\\|^2=p$). Models for which an asymptotic characterization of such GU breaking are available and studied. The design of new, more sophisticated frameworks in which GU breaks down might be of technical interest but nevertheless of limited general relevance, unless they lead to a precise and general characterizing result. For these reasons, I am therefore inclined to keep my score as it is.
> > >
> > > As a side note, about my question on the ridge regularization and the asymptotic estimator characterization, what I was referring to is that in case, for example, of $L_1$ loss, the explicit expression in Eq. 16 would not be possible in my understanding. This is however a marginal aspect.

---

> ### Author Response · Authors · 2024-11-24
>
> Thanks to the reviewer for the feedback.
>
>
> While we cannot argue with the reviewer on the "general relevance" of this work as it is reserved to personal judgement, we would like to point out that **precise and general** characterizing results are particularly hard to obtain in the high-dimensional asymptotic regime, to our knowledge such results are scarce (if not absent) in the literature. We would appreciate it if the reviewer could provide some references that contain characterizing results deemed to be **precise and general** by the reviewer.
>
> The reviewer seems to concur that this paper provides a **novel** analysis in the high-dimensional asymptotic regime. In our opinion, this fact alone (combined with the popularity of linear factor models) already qualifies this work as a significant contribution in the field of high-dimensional asymptotic analysis (which we argue to be an important field but still **unfamiliar** to many of the audience and reviewers of ICLR). We kindly refer the reviewer to General Reply to all Reviewers for further comments on the significance of this work.
>
>
> We respectfully do not agree that this work provides little further insight into the breakdown of Gaussian universality beyond existing results. At the risk of repeating ourselves, we would like to point out the point-wise normality condition is **not** directly verifiable from the data distribution. Therefore, it provides limited insight into whether the Gaussian universality holds under a given data model. As the reviewer might also notice, the counterexample provided in [Montanari & Saeed (2022)](https://arxiv.org/pdf/2202.08832) is 'manufactured' with a **particular loss function** to induce the breakdown of Gaussian universality on Radermacher data $\mathbf{x}={\rm Unif}\\{-1,1\\}$. This counterexample clearly does **not** mean that the Gaussian universality cannot hold on $\mathbf{x}={\rm Unif} \\{-1,1\\}$ for a different loss function (see for instance Figure 2 in [Taheri et al. (2021)](https://www.mdpi.com/1099-4300/23/2/178) for **universality results on Radermacher data**). In our view, the existence of such designed counterexamples in the literature attest to the interest in the research community to understand better when the Gaussian universality holds and when it fails, which supports the value of the present work, rather than undermines it.

---

> > ### Comment · Reviewer_8nd6 · 2024-11-24
> >
> > Thanks to the authors for their prompt reply. What I meant by referring to the "precise and general" results is not that such a result in the literature exists but that, ideally, such a result would be the desirable step forward with respect to a set of investigations that have shown quantitatively a breakdown of GUs in the asymptotic regime.
> >
> > I hope not to be misunderstood on the fact that I think this is a nice contribution and I do not claim that there is *no novelty* in it. However, my (as I said, *mildly*) negative rating is due to the fact that such novelty seems to me essentially focused on the adoption of LFMM, rather than the *type* of results and described phenomenology announced in the abstract (asymptotic characterization of the model, GU breaking, optimality of the loss which is GU-breaking dependent).
> >
> > In particular, with respect to the list of main contributions given in the paper, the authors refer, in their first point, to the fact that they ```provide in Theorem 1 an asymptotic characterization of ridge-regularized empirical risk
> > minimization (ERM) for the classification of data drawn from a linear factor mixture model [...]. This precise characterization gives access to the asymptotic performance on mixture data beyond Gaussian universality```.  This kind of precise characterization is of the type given in the aforementioned works of [El Karoui et al.](https://www.pnas.org/doi/10.1073/pnas.1307842110) (who used the same leave-one-out method) and, specifically for the case of classification, [Adomaityte et al.](https://proceedings.neurips.cc/paper_files/paper/2023/hash/88be023075a5a3ff3dc3b5d26623fa22-Abstract-Conference.html) (via the replica method). The paper of El Karoui and coworkers does not focus on GU as, at the time, GU was not the spotlight as it is now and therefore this was not stressed as much. The third point the authors give as the main contribution is the fact that square loss optimality does not hold anymore for LFMM due to the breakdown of Gaussian universality. The dependence of the optimal loss function in the presence of Gaussian universality breaking was observed again by [Adomaityte et al. (2023)](https://proceedings.neurips.cc/paper_files/paper/2023/hash/88be023075a5a3ff3dc3b5d26623fa22-Abstract-Conference.html) for classification, with respect to the optimal performance analyzed in the Gaussian case by [Mignacco et al. (2020)](https://proceedings.mlr.press/v119/mignacco20a.html), and, in a second paper, by [Adomaityte et al. (2023)](https://arxiv.org/pdf/2309.16476) for regression, even in the case in which, in my understanding, the finiteness of the moments of $a$ guarantees a concentration of the quadratic forms, but show that asymptotic characterization in this proportional regime is possible, GUs appears to be broken and optimality results for the Gaussian case with respect to the loss choice might need to be revisited. Of course, these results cannot be absolutely identical in the present setting, as the model is different. The interesting feature of the considered model is (to me) what the authors mention in the second point of their first reply above, and in the second point of their *Our contributions* section, i.e., the fact that the model can specifically be analyzed so that, under some condition, GU *does* hold. Nevertheless, my personal impression is that the contribution is more of an interesting *variation* on the theme of models that are asymptotically analyzable and exemplify how GU-based results can be fragile.
> >
> > Finally, I agree that the counterexample provided by Montanari and Saeed is manufactured, and I mentioned it just to refer to a simple, known case in which the concentration of the norm of the covariates holds (unlike the elliptical distribution case), yet GU is not realized.

---

> ### Author Response · Authors · 2024-11-25
>
> We thank the reviewer for acknowledging that the present work is "a nice contribution" of adequate "novelty", and for providing concrete comments that help us better understand the standpoint of the reviewer.
>
> Thanks also for the confirmation that "general and precise" results are out of reach in the current literature.
> We would like to emphasize that it is **not** our intention to claim that our analysis provides a major breakthrough that puts it on a different level from other important contributions in the field (such as the ones mentioned by the reviewer). The point that we are trying to make is simply that it contains sufficiently novel and interesting results to be a worthy publication with respect to previous related works.
>
> As mentioned by the reviewer, high-dimensional asymptotic analyses for elliptical data, such as [Adomaityte et al. (2023)](https://proceedings.neurips.cc/paper_files/paper/2023/hash/88be023075a5a3ff3dc3b5d26623fa22-Abstract-Conference.html), [Adomaityte et al. (2024)](https://arxiv.org/pdf/2309.16476), and [El Karoui et al. (2013)](https://www.pnas.org/doi/pdf/10.1073/pnas.1307842110), are important contributions in the field even without considering their implication on Gaussian universality. In our opinion, this is also the case for our work: the **novelty of the analysis** and the **importance of linear factors models** under study already provide strong arguments on the significance of this paper.
>
> We would like to stress again that the purpose of our results is not to disprove Gaussian universality or to show to which point "Gaussian universality results can be fragile", but to better understand **when this important phenomenon can be expected to hold**, which is a crucial question only **partially answered** by existing results such as the point-wise normality condition and the manufactured counterexample in [Montanari & Saeed (2022)](https://arxiv.org/pdf/2202.08832). As correctly understood by the reviewer, studying Gaussian universality under LFMM has the particular interest of revealing conditions of Gaussian universality breakdown. Note also that these conditions are easy to verify from the (Gaussian or non-Gaussian) distributions of informative factors and the type (square or non-square) of loss function used in ERM. See the numerical results in Section 5 and Appendix B on synthetic and real data, where we can **predict** the GU behavior **before** running the ERM.
> Another important point we would like to highlight (again) is that linear factor models are among the **most basic probabilistic models** [(Goodfellow et al., 2016, Chapter 13))](https://www.deeplearningbook.org). They are, in our view, **not less important** than elliptical data models, and are by no means particular data models intentionally designed by the authors to induce the Gaussian universality breakdown.
>
> As briefly explained in our first reply to the reviewer, elliptical data vectors $\mathbf{x}=a\mathbf{u}$ with **random scalar** $a$ and **Gaussian vector** $\mathbf{u}\sim\mathcal{N}(\mathbf{0},\mathbf{I}_p/p)$ considered in [Adomaityte et al. (2023)](https://proceedings.neurips.cc/paper_files/paper/2023/hash/88be023075a5a3ff3dc3b5d26623fa22-Abstract-Conference.html), [Adomaityte et al. (2024)](https://arxiv.org/pdf/2309.16476), and [El Karoui et al. (2013)](https://www.pnas.org/doi/pdf/10.1073/pnas.1307842110) are **not of concentrated norm**. To see this, notice first that $\Vert\mathbf{u}\Vert^2=1+o(1)$ at large dimension $p$ by the law of large numbers. Consequently $\Vert\mathbf{x}\Vert^2=a^2\Vert\mathbf{u}\Vert^2=a^2+o(1)$, which obviously does **not concentrate around a constant** as long as $a$ is not constant. Therefore the finiteness of the moments of $a$ **does not guarantee** a concentrated norm of $\mathbf{x}$. Another major difference (not mentioned in the first reply) between elliptical data vectors $\mathbf{x}=a\mathbf{u}$ considered in these works and our LFMM is that even though $\mathbf{x}=a\mathbf{u}$ is clearly non-Gaussian, it follows a **conditional Gaussian distribution** (as the distribution of $\mathbf{x}$ conditioned on $a$ is a multivariate normal distribution), which is **not the case under our LFMM**.
>
> Finally we still do not see the problem with the first point of our contributions quoted by the reviewer. To us, the fact that this analysis focuses on "classification of data drawn from a linear factor mixture model" is clearly stated. And in no part of this description did we ever attempt to exclude the existence of other non-universal results in the literature. In fact, all the references mentioned by the reviewer had been cited and discussed in our submission. We also adapted remarks on these works in the revised version after exchanging with the reviewer. We would be grateful if the reviewer could provide any further suggestion or question on our discussion of related works.

---

> > ### Comment · Reviewer_8nd6 · 2024-12-02
> >
> > I would like to thank the authors for their reply. As per my "general" reply above, I think it is important to stress that this contribution is not the first one providing an asymptotic characterization of GU breaking in the context of classification, and a number of results in this sense have been already collected concerning both the breaking per se and the change in loss optimality induced by fat tails, but nevertheless investigate a *different* relevant setup that can complement such collection of findings. As the authors already agreed to do, I recommend a specification in the title of the considered setup, and possibly a clearer specification that the characterization of asymptotic GU universality breaking is within the contributions given by previous authors, although for different models. As the authors agreed to incorporate these suggestions, which I think are important to not mislead the reader, I am increasing my score.

---

### Official Review · Reviewer_hVpL · 2024-11-01

**Soundness:** 3
**Presentation:** 3
**Contribution:** 2
**Rating:** 6
**Confidence:** 3

**Summary:**

The paper studies on the issue of 'Gaussian universality' for classification of high dimensional data drawn from a Linear Factor Model (LFMM) that constitutes a GMM generalization. It focuses on LFMM classifiers trained using squared loss and ridge regression. The main theoretical result is that, if LFMMs with non-Gaussian informative factors are used, 'Gaussian universality' breaks down.

**Strengths:**

S1. The paper presents theoretical results related to the Gaussian Universality of classification mixtures for high dimensional data.

S2. It identifies a case namely, LFMMs with non-Gaussian  informative factors trained using squared loss ridge regression, where Gaussian universality breaks.

S3. Rigorous proofs are provided.

**Weaknesses:**

W1. Although the paper is well-structured, it is rather difficult to follow due to its theoretical character.
Nevertheless, I think that the main theoretical conclusions could be explained in a more comprehensive way.

W2. The authors should also comment on the importance of the presented results from a practical point of view.

**Questions:**

Q1. See comment W2 above.

Q2. The paper proves a negative theoretical result regarding the squared loss, however a suggestion on the type of loss that could
be considered would add value to the paper.

---

> ### Author Response · Authors · 2024-11-20
>
> We thank the reviewer for the valuable comments.
>
> We would like to clarify that our analysis is **not limited to the square loss**, but instead applies to a family of **smooth convex losses** (see Assumption 1 for more details).
>
> **Comment on practical relevance**
>
> The research line of high-dimensional asymptotic analyses of ML methods such as ours is gaining interest and gives rise to a rapidly growing volume of publications (see for instance the numerous references cited in the introduction of the manuscript).
> It is driven by the need for a better understanding of modern large-scale ML with comparably large feature dimension $p$ and sample size $n$, where the practical reality often contradicts the classical learning theory according to which overfitted large learning models like deep neural networks should generalize poorly on newly observed data instances.
> As high-dimensional asymptotic analyses give the **exact** generalization performance at **any finite** sample ratios $n/p$, they allow a deep understanding of the learning mechanism and provide practical guidance in the modern big data regime.
>
> Due to the technical challenge of characterizing the exact performance at finite $n/p$, *most* analyses in the literature considered Gaussian features or/and used arguments of Gaussian universality. The practical insights provided in these works rely on how much the Gaussian setting is able to describe the practical reality. In comparison, our work
>
> - sheds light into the interplay between the learning model and the data statistics **beyond** the first and second moments, through a precise performance analysis **beyond** the realm of Gaussian universality; and
> - provides guidance on when real data can be expected to behave as Gaussian data, based on the identified conditions of Gaussian universality under LFMM; and
> - lays foundation for future works on the optimal loss design better adapted to non-Gaussian features.
>
> **Comment on optimal loss**
>
> High-dimensional asymptotic analyses typically give an implicit characterization of the generalization performance through a system of equations, making it challenging to find the optimal loss (that potentially depends not only on the data distribution but also on the sample ratio $n/p$). For that reason, investigations on the optimal loss in the high-dimensional asymptotic regime often give rise to independent publications, for instance [1, 2]. While our precise high-dimensional analysis can be used to design optimal loss adapted to non-Gaussian features under LFMM, such results are of independent interest and beyond the focus of this paper, which, in our opinion, already contains sufficient novel results.
>
> [1] Bean, D., Bickel, P.J., El Karoui, N. and Yu, B., 2013. Optimal M-estimation in high-dimensional regression. Proceedings of the National Academy of Sciences, 110(36), pp.14563-14568.
>
> [2] Taheri, H., Pedarsani, R. and Thrampoulidis, C., 2021. Sharp guarantees and optimal performance for inference in binary and Gaussian-mixture models. Entropy, 23(2), p.178.

---

> ### Comment · Reviewer_hVpL · 2024-11-26
>
> I thank the authors for their detailed reply to all reviewers. The paper contains interesting results, however I am not confident that the importance as well as the relevance of those results justify an ICLR publication.

---

> > ### Author Response · Authors · 2024-11-26
> >
> > Thanks to the reviewer for acknowledging the interest of our results. While the "importance" and "relevance" of the results is certantly a legitimate criterion, we would appreciate if the reviewer could be more specific about it than generic comments such as not enough to "justify a ICLR publication".
> > At the risk of stating the obvious, the reviewer is welcome to raise any question or concern on the novelty and significance of our results **in comparison** to any related work (published in "top venues" judged by the reviewer, a standard which we believe is met by many of the references cited and discussed in our paper).

---

> > > ### Comment · Reviewer_hVpL · 2024-12-02
> > >
> > > Based on the extensive discussion among authors and reviewers I will increase my score.

---

### Official Review · Reviewer_LD77 · 2024-11-04

**Soundness:** 4
**Presentation:** 4
**Contribution:** 2
**Rating:** 3
**Confidence:** 3

**Summary:**

Paper shows that a mixture of linear factors model with non-Gaussian noise doesn't satisfy "Gaussian universality" in terms of asymptotic analysis. This is what I'd call a negative result, and the authors argue this is significant because previously researchers have shown that in various cases you can get away with swapping in Gaussians when doing asymptotic analysis using some central limit theorem arguments.

**Strengths:**

Focus: There focus on mixtures of linear factor models is quite reasonable and practical and theoretically satisfying in that they generalize GMMs.

Correctness: I didn't see any problems in the math and it all smelled clean, but I didn't fine tooth comb.

This seems like a perfectly nice bit of results that I'd like to see published somewhere someday, but I just can't personally argue that it meets the high bar for ICLR in terms of significance.

**Weaknesses:**

They have to make a non-Gaussian noise assumption, and then the number of non-Gaussian noise variables goes to infinity whereas the number of informative features $q$ stays fixed. Now CLT says that even in the presence of a lot of non-Gaussian noise one can end up with Gaussians, so some might argue it's still surprising that this failed a CLT like story, but this isn't the usual CLT set-up, so perhaps not so surprising that infinite non-Gaussian noise leads to non-Gaussian-swappable asymptotics.

**Questions:**

The authors declare that they have shown squared error isn't an optimal loss *in this setting* and state that as an important contribution. Why is that surprising though, I mean it's easy to conjure up scenarios where that's true (e.g. Laplacian noise -> L1 loss).  So why should we be surprised if you assume non-Gaussian noise on your features and then tell us squared loss is not optimal?

---

> ### Author Response · Authors · 2024-11-20
>
> We thank the reviewer for the valuable comments.
>
> **Significance of this work**
>
> We would like to stress that the main objective of this paper is **not to provide a negative result** to disprove the Gaussian universality principle, but rather to provide a **precise performance analysis of classification under linear factor mixture models** (LFMM, which, to quote the reviewer, is a `quite reasonable and practical and theoretically satisfying` data setting), and then to identify conditions of Gaussian universality breakdown under LFMM, which shed light on the limitation of the extensively exploited Gaussian universality.
>
> Please see the general reply to all reviewers above for more detailed comments on these points.
>
> **Comment on the sub-optimality of square loss**
>
> We did not intend to point out the sub-optimality of square loss as ''*an important contribution*'' of this work, but as an argument on the interest of our LFMM data setting, compared to the commonly considered GMM setting, in capturing key aspects of learning behavior, such as the impact of loss function.
>
> In the modern high-dimensional regime where the sample size $n$ and the feature dimension $p$ are both large (as considered in this paper, which we argue to be more practical for modern large-scale ML, see the first paragraph of introduction for references), it is known that the maximum likelihood estimator can be sub-optimal and the choice of optimal loss is much less clear. See for instance Figure 1 of [1], where the optimal loss is shown to be **different from the $L_1$ loss** in the Laplacian case at finite $n/p$, as opposed to the classical result of maximum likelihood theory mentioned by the reviewer.
> In fact, the classical maximum likelihood theory often fails in the large $n,p$ regime, as evidenced by the findings of [2].
> Another example is given in Section 3.2 of [3], where it was shown that in this high-dimensional regime, the optimal loss is **not** the square loss, even when the features are Gaussian distributed (e.g., for logistic and probit models).
>
>
> Even though precise performance analyses such as ours allow access to the generalization performance as a function of the loss function and the sample ratio $n/p$, the asymptotic generalization performance is usually expressed in an implicit form involving a system of equations.
> Optimizing the generalization performance over the choice of loss function is hence a non-trivial task, often warranting separate investigations such as [1, 3].
> One contribution of our analysis is laying the ground for future works on the search for optimal loss under LFMM.
>
> [1] Bean, D., Bickel, P.J., El Karoui, N. and Yu, B., 2013. Optimal M-estimation in high-dimensional regression. Proceedings of the National Academy of Sciences, 110(36), pp.14563-14568.
>
> [2] Sur, P. and Candès, E.J., 2019. A modern maximum-likelihood theory for high-dimensional logistic regression. Proceedings of the National Academy of Sciences, 116(29), pp.14516-14525.
>
> [3] Taheri, H., Pedarsani, R. and Thrampoulidis, C., 2021. Sharp guarantees and optimal performance for inference in binary and Gaussian-mixture models. Entropy, 23(2), p.178.

---

> > ### Comment · Reviewer_LD77 · 2024-11-24
> > **enjoyed authors response**
> >
> > I thank the authors for the interesting and insightful response to my review. I have also read the other reviewers comments and author discussion, and found they had some useful points as well. I especially found the discussion of Adomaityte et al. (2024) helpful, and hope the authors will include similar clear explanations in the final version of this manuscript, and also the discussion of the counterexample provided in Montanari & Saeed (2022) and why this work offers something new.   In general, I have high respect for this work and agree it adds something to this vein, but I can't argue it provides significant enough new insights or results for ICLR.

---

> > > ### Author Response · Authors · 2024-11-24
> > >
> > > Our thanks to the reviewer for the positive feedback to our reply, and we are working to include the discussions and additional experiments suggested by the reviewers in an updated version of the manuscript.
> > > To avoid burdening the discussion page with redundant remarks, we kindly refer the reviewer to our last reply to Reviewer 8nd6 for our comments on the counterexample provided in [Montanari & Saeed (2022)](https://arxiv.org/pdf/2202.08832).
> > > To better address the persistent concern of the reviewer on the significance of this work, and to further improve this paper, it would be helpful if the reviewer could be more precise about what is lacking in the present work with respect to previous publications in the field of high-dimensional asymptotic analysis. In particular, we would appreciate it if the reviewer could point us to previous works where similar results and/or insights to ours were given, so that we can provide further theoretical and/or empirical comparisons.

---

### Author Response · Authors · 2024-11-20
**General reply to Reviewers**

We thank all Reviewers for their time and effort.
After reading your feedback, we would like to address here a common concern on the  **significance** of this work.
In our view, the present contribution constitutes a meaningful addition to the long list of high-dimensional asymptotic analyses of ML methods in the literature, in that
- it characterizes the **exact classification performance** on mixture data drawn from **linear factor models**, which, despite being a basic probabilistic framework in statistical inference and generative models, have **not** yet been studied before under the high-dimensional asymptotic regime;
- it reveals a system of equations controlling the classification performance that is **different** from those obtained under Gaussian mixture models (GMMs), therefore **unattainable** through arguments of Gaussian universality (that have been exploited in many previous efforts on high-dimensional asymptotic analyses of ML methods);
- by deriving conditions under which the system of equations is reduced to the one that holds for GMMs, it allows deeper insights into when Gaussian universality can be expected to hold in classification of mixture data beyond the conditional one-directional CLT recently discovered by Dandi et al. (2024).

Other questions and concerns are addressed in individual responses to the reviewers.

The modifications made in the revised version are marked in red.

---

> ### Author Response · Authors · 2024-11-25
> **Second general reply to Reviewers**
>
> We would like to thank the reviewers for recognizing the quality of our reply (even though not yet reflected in the scores).
> To test the practical applicability of our results on real data (as per the suggestion of Reviewer 66dx), we added, in Appendix B of the (newest) revised manuscript, additional experiments on Fashion-MNIST image data, where we showed how our universality conditions on the (Gaussian or non-Gaussian) distributions of informative factors and the type (square or non-square) of loss stated in Corollaries 2 and 3 can be used to **predict** Gaussian universality phenomena (or their breakdown) with respect to the in-distribution performance and the ERM classifier (see Definition 3 for more details on these two types of Gaussian universality).

---

### Author Response · Authors · 2024-12-02
**Message to Reviewers**

Dear Reviewers,

Thanks again to you all for your replies and the unanimous acknowledgment of the quality and interest of our results. The only remaining concern appears to be the significance level of the present work, which, in our understanding, should be largely based on its positioning w.r.t. the literature.
On that point, we would like to thank Reviewer 8nd6 for the insightful questions that help us to better position our work, and Reviewer 66dx for the suggestion to illustrate the practical interest of our theoretical results on real data. With the discussion period approaching the end (on **December 2nd**), we would like to make a summary of the main points that we believe to be important to the assessment of the significance of this work.

**Significance of precise analyses under non-Gaussian data models**

As mentioned by Reviewer 8nd6, some previous publications in prestigious venues such as [Adomaityte et al. (2024)]( https://proceedings.neurips.cc/paper_files/paper/2023/hash/88be023075a5a3ff3dc3b5d26623fa22-Abstract-Conference.html), [El Karoui et al. (2013)](https://www.pnas.org/doi/pdf/10.1073/pnas.1307842110) focused on precise performance analyses under elliptical data models without specially mentioning Gaussian universality.
Indeed, the main contribution of the very recent publication [Adomaityte et al. (2024)]( https://proceedings.neurips.cc/paper_files/paper/2023/hash/88be023075a5a3ff3dc3b5d26623fa22-Abstract-Conference.html) is a precise understanding of classification on high-dimensional mixtures of elliptical data. It goes without saying that other important data models in statistical learning besides elliptical models deserve the same attention from the research community. The significance of our contribution is justified from that angle alone: it is the fist to characterize the exact classification performance on high-dimensional mixture data drawn from **linear factor models** -- a fundamental probabilistic framework **not less important than elliptical models**, see [(Goodfellow et al., 2016, Chapter 13))](https://www.deeplearningbook.org).

**Significance of understanding the impact of non-Gaussian data beyond the data vector norms**

As pointed out by Reviewer 8nd6, the previous works [Adomaityte et al. (2024)](https://proceedings.neurips.cc/paper_files/paper/2023/hash/88be023075a5a3ff3dc3b5d26623fa22-Abstract-Conference.html), [Adomaityte et al. (2024)](https://arxiv.org/pdf/2309.16476), and [El Karoui et al. (2013)](https://www.pnas.org/doi/pdf/10.1073/pnas.1307842110) also shed light on the impact of non-Gaussian data. However, as the elliptical data vector $\mathbf{x}=a\mathbf{u}$ considered in these works is a Gaussian vector $\mathbf{u}\sim\mathcal{N}(\mathbf{0},\mathbf{I}_p/p)$ rescaled by a random factor $a$, these analyses *do not* provide further information besides the impact of the data vector norms controlled by the scaling factor $a$. For instance, they do not apply to the classical case of i.i.d. non-Gaussian features, which in fact corresponds to a linear factor model with identity covariance.

**Significance of verifiable conditions of Gaussian universality**

Many newest results on Gaussian universality (GU) focused on the pointwise normality conditions (see a list of major contributions in Section 2). While revealing a key element of GU, these conditions are difficult to check, as they require knowledge on the statistical behavior of the ERM solution, which is in general *inaccessible* without precise analysis. In other words, given a data distribution and a loss function, the pointwise normality conditions cannot be used to predict GU, except in some special cases like the manufactured counterexample of [Montanari & Saeed (2022)](https://arxiv.org/pdf/2202.08832) mentioned by Reviewer 8nd6.
While this counterexample showed the breakdown of GU on Rademacher data for a **particularly designed** loss function, there was *no* theoretical conclusion on whether GU can hold for other losses. In fact, for several popular losses, empirical observations of universal performances on Rademacher data were reported in Figure 1&2 of [Taheri et al. (2021)](https://www.mdpi.com/1099-4300/23/2/178).

In contrast, our conditions of GU *are easy to verify* by simply answering whether the informative factors are exclusively Gaussian and whether the square loss is used. As evidenced by our experiments on real data in Appendix B of the revised manuscript, we can predict, from the yes-or-no answers to these questions, the universal or non-universal behavior on Fashion-MNIST image data w.r.t. the in-distribution performance and the classifier without running the ERM algorithm or conducting the precise analysis.

We hope that this summary is concise and informative. If you find these points on significance satisfying, we kindly ask that you increase your score. If you have any further question or\and comment, we will be more than happy to address them.

Best,

Authors

---

> ### Comment · Reviewer_8nd6 · 2024-12-02
>
> I would like to thank the authors for the summary. As a side note that came to my mind by reading it, the authors might consider stressing the originality of the contribution regarding **linear factor models** with respect to previous papers dealing with the breaking of GU in the classification of mixtures, by referring to LFMM in the title, e.g., "The Breakdown of Gaussian Universality in Classification of High-dimensional Linear Factor Mixtures" or something similar.

---

> > ### Author Response · Authors · 2024-12-02
> > **Reply to Reviewer 8nd6**
> >
> > We thank Reviewer 8nd6 for the very good suggestion about changing the title. We will modify the title as suggested in the final version. If the reviewer is satisfied with our arguments on the significance and originality of this work, please remember to increase the scores accordingly.

---

> > > ### Author Response · Authors · 2024-12-02
> > >
> > > We thank Reviewer 8nd6 and Reviewer hVpL for their prompt feedbacks and for increasing their scores. With the discussion period ending in less a day, we would like to invite Reviewer LD77 to take a look at the above summary of important points on the significance and originality of our work w.r.t. the major contributions in the field, and to adjust the rating accordingly if our arguments adequately address the reviewer's remaining concern on significance.

---

> > > > ### Comment · Reviewer_LD77 · 2024-12-03
> > > > **Acknowledging Authors comments**
> > > >
> > > > Hi! Yea I have read all the discussion to date, and I still don't think this is significant enough for ICLR, but if the Area Chair feels it's an important contribution, I'm not against accepting this paper, I just can't argue for accepting it.

---

> ### Author Response · Authors · 2024-12-03
> **Message to Reviewer LD77**
>
> Dear Reviewer LD77,
>
> It is our understanding that review ratings should be based on concrete arguments. We looked again in your replies and failed to identify one, other than the repeated generic comment about not meeting the “bar” for ICLR. Good arguments to support such criticisms could be, for instance, lack of originality or interest w.r.t. the previous works. Our message above addresses precisely these points. If you have further question or comment, please let us know before the discussion deadline.
>
> It is also our belief that your rating should reflect faithfully your current stand on the paper. If you are just uncertain about this paper as you said in your last reply, we kindly ask you to at least increase your score to borderline and to drop your confidence level.

---

### Meta-Review · Area_Chair_mLRj · 2024-12-21

**Metareview:**

(a) The paper, "The Breakdown of Gaussian Universality in Classification of High-dimensional Mixtures," investigates the limitations of Gaussian universality (GU) in high-dimensional classification tasks. The authors provide a precise asymptotic characterization of empirical risk minimization (ERM) under linear factor mixture models (LFMM), a generalization of Gaussian mixture models (GMMs). They show that the Gaussian universality principle breaks down in this setting, with learning performance depending on higher-order moments of the data distribution. Additionally, the paper derives conditions under which GU holds and discusses implications for choosing loss functions.

(b) **Strengths:**
   - Rigorous mathematical analysis using leave-one-out techniques.
   - Precise characterization of performance in a high-dimensional regime, advancing the understanding of GU limitations.
   - Novel insights into when GU can be expected to hold or break under LFMM.
   - Contributions to the understanding of the interplay between data distribution and loss function design.

(c) **Weaknesses:**
   - The paper's scope may seem incremental compared to existing studies, such as those analyzing elliptical data models.
   - The practical implications of the theoretical results could be more clearly emphasized, particularly in real-world scenarios.
   - Some results, such as the non-optimality of the squared loss under certain conditions, are not unexpected and could benefit from further elaboration.

(d) **Reasons for acceptance:**
   The paper offers significant theoretical contributions to the understanding of GU in high-dimensional settings, a critical area in modern machine learning. Its rigorous methodology and clear exposition provide a valuable resource for further research. The authors' identification of GU conditions and their implications enrich the literature. While some reviewers expressed concerns about the work's significance, the combination of novelty, soundness, and relevance to the ICLR audience justifies acceptance.

---

**Additional Comments On Reviewer Discussion:**

The review process for this paper involved active engagement between the authors and reviewers, with substantial clarifications provided during rebuttal. Key points raised include:

- **Reviewer LD77's concerns**: The reviewer acknowledged the quality of the work but questioned its significance relative to the ICLR bar. Despite the detailed rebuttal, they maintained a lower score due to subjective concerns about impact rather than methodological flaws.

- **Reviewer hVpL and 8nd6**: These reviewers initially raised issues about clarity and practical relevance. The authors addressed these with detailed responses, including added experiments on real datasets and contextualization of their work with prior literature. Both reviewers subsequently increased their scores.

- **Strengths acknowledged**: Rigorous proofs, novel model analysis, and theoretical contributions were uniformly appreciated. The concerns centered on the broader impact and incremental nature of the findings compared to existing works on GU.

**Final decision considerations**:
While not all reviewers were fully convinced of the work's transformative impact, the methodological rigor, clarity of exposition, and valuable theoretical insights weigh heavily in favor of acceptance. The authors have adequately addressed most concerns, reinforcing the paper's contribution to the field.

---

### Decision · Program_Chairs · 2025-01-22

Accept (Poster)